# The CaMKII/NMDA receptor complex controls hippocampal synaptic transmission by kinase-dependent and independent mechanisms

Salvatore Incontro[1], Javier Díaz-Alonso[1], Jillian Iafrati[2], Marta Vieira[3], Cedric S. Asensio [4], Vikaas S. Sohal[2], Katherine W. Roche[3], Kevin J. Bender[5] & Roger A. Nicoll[1,6]

CaMKII is one of the most studied synaptic proteins, but many critical issues regarding its role in synaptic function remain unresolved. Using a CRISPR-based system to delete CaMKII and replace it with mutated forms in single neurons, we have rigorously addressed its various synaptic roles. In brief, basal AMPAR and NMDAR synaptic transmission both require CaMKIIα, but not CaMKIIβ, indicating that, even in the adult, synaptic transmission is determined by the ongoing action of CaMKIIα. While AMPAR transmission requires kinase activity, NMDAR transmission does not, implying a scaffolding role for the CaMKII protein instead. LTP is abolished in the absence of CaMKIIα and/or CaMKIIβ and with an autophosphorylation impaired CaMKIIα (T286A). With the exception of NMDAR synaptic currents, all aspects of CaMKIIα signaling examined require binding to the NMDAR, emphasizing the essential role of this receptor as a master synaptic signaling hub.

[1] Department of Cellular and Molecular Pharmacology, University of California, San Francisco, San Francisco, CA 94158, USA. [2] Department of Psychiatry, University of California, San Francisco, San Francisco, CA 94158, USA. [3] Receptor Biology Section, National Institute of Neurological Disorders and Stroke, National Institutes of Health, Bethesda, MD 20892, USA. [4] Department of Biological Sciences, University of Denver, Denver, CO 80208, USA. [5] Department of Neurology, University of California, San Francisco, San Francisco, CA 94158, USA. [6] Department of Physiology, University of California, San Francisco, San Francisco, CA 94158, USA. Correspondence and requests for materials should be addressed to S.I. (email: salvincontro@gmail.com) or to R.A.N. (email: roger.nicoll@ucsf.edu)

Calcium-calmodulin-dependent protein kinase II (CaMKII), one of the first proteins identified in the postsynaptic density (PSD), has remained a focus of intense research ever since[1–4]. In the forebrain CaMKII consists of heteromers of alpha and beta subunits, with an excess of CaMKIIα compared to CaMKIIβ[5]. The protein is present in unusually high amounts for a kinase, which led to an early hypothesis that it also has a structural role. CaMKII is inactive under resting conditions, as substrate access to its binding site in the catalytic domain is blocked by the autoinhibitory pseudosubstrate of the protein[4,6,7].

$Ca^{2+}$ influx through synaptic N-metil-D-aspartate receptors (NMDARs) binds to calmodulin (CaM), which then binds to the pseudosubstrate segment of CaMKII, relieving autoinhibition[1,6,8]. When $Ca^{2+}$/CaM binds, autophosphorylation of CaMKIIα at T286 results in kinase activity that persists after removal of $Ca^{2+}$/CaM[6,7]. These properties have made CaMKII an extremely popular molecular model for information storage. Indeed, both pharmacological blockade of CaMKII[9–11] and genetic deletion of CaMKII[12–14] strongly reduce NMDAR-dependent long-term potentiation (LTP), but rarely eliminate it, raising the possibility

of a CaMKII-independent component to LTP. Expression of exogenous constitutively active CaMKII closely mimics LTP[15–17], suggesting that it is sufficient for LTP.

Most studies have focused on CaMKII's enzymatic role at excitatory synapses; however, kinase-independent structural roles for CaMKII have recently emerged[3]. These structural roles appear to depend on CaMKIIβ, which localizes to the post-synaptic density (PSD) through interactions with F-actin[18–20]. For instance, the morphological effects of deleting CaMKIIβ can be rescued by expressing a kinase dead mutant of CaMKIIβ[20]. In addition, the impairment of CaMKIIα targeting to the PSD in the CaMKIIβ KO mouse is not observed in a knockin mouse expressing the Thr286 autophosphorylation null CaMKIIα mutant (T286A)[21]. Thus it has been postulated that the two CaMKII subunits serve separate roles, with CaMKIIα being primarily recruited to the PSD in an activity-dependent manner during LTP[1,2], whereas CaMKIIβ stabilizes the actin cytoskeleton.

Given these dual roles, one would suspect that CaMKII contributes both to changes in synaptic strength as well as basal synaptic transmission. Nevertheless, the effect, if any, that CaMKII has on basal synaptic transmission is confusing. Pharmacological blockade of CaMKII has mixed effects on baseline transmission[9,10,22–25], whereas complete deletion of CaMKIIα either in the germline KO[12] or in the adult conditional KO[26] has no effect on basal transmission. Furthermore, while knockin of the Thr286 autophosphorylation null CaMKIIα mutant[13] or a kinase dead mutant[14] inhibits LTP, it does not alter baseline transmission. Overall, these results suggest that CaMKII is not required for normal synapse development or basal synaptic strength. Rather it is specifically dedicated to LTP.

It is clear that while CaMKII has remained a central focus of studies on synaptic plasticity for over two decades, many unresolved issues remain. In the present study we have used a CRISPR-based system to address many of these issues. Deleting CaMKIIα acutely caused a dramatic reduction in α-amino-3-hydroxy-5-methyl-4-isoxazolepropionic acid receptor (AMPAR) excitatory post-synaptic currents (EPSCs) and a modest reduction in NMDAR EPSCs. Replacement of wild-type (WT) CaMKIIα with autophosphorylation impaired CaMKIIα T286A and kinase impaired CaMKIIα K42R failed to rescue the AMPAR defect, but did rescue the NMDAR defect, indicating that maintenance of basal AMPAR transmission, but not NMDAR transmission, requires CaMKIIα activity. The blockade of LTP by deleting CaMKIIα was not secondary to the reduction in NMDAR EPSCs, because when NMDAR currents are rescued by replacing endogenous CaMKIIα with mutated forms, LTP was still prevented. Our findings have clearly delineated both enzymatic and structural roles for CaMKII in maintaining basal synaptic transmission

in addition to its essential role in synaptic plasticity. Finally, disrupting the binding of CaMKII to NMDARs abolishes all examined actions of CaMKII, except its ability to rescue NMDAR synaptic currents. Our results demonstrate the unappreciated role of CaMKII in basal transmission and clarify the literature by delineating the relative contributions of CaMKIIα and CaMKIIβ in our KO system. We demonstrate the central role of the CaMKII/NMDAR protein complex as a key-signaling hub, controlling numerous fundamental aspects of excitatory synaptic transmission.

## Results

**The role of CaMKII in basal synaptic transmission and LTP.** We first designed a number of guide RNAs (gRNAs) against CaMKIIα and CaMKIIβ. To test their effectiveness we expressed these gRNAs together with Cas9 (Fig. 1a) with lentivirus in dissociated neuronal cultures. We identified two gRNAs that were highly effective in eliminating CaMKIIα protein (Fig. 1b) and one gRNA highly effective against CaMKIIβ (Fig. 1b). We also verified these gRNAs with immunofluorescence in dissociated neuronal cultures and found little detectable protein for either CaMKIIα (Fig. 1c−e) or CaMKIIβ (Fig. 1f−h) in transfected cells. Next, we coated gold particles with Cas9 and the gRNA, biolistically delivered them to hippocampal slice cultures and waited 10 −12 days before recording (Fig. 1a, bottom) (see Methods). Simultaneous paired recordings were made from a transfected cell and a neighboring control cell, while stimulating a common population of excitatory afferents (Fig. 1a, bottom). AMPAR EPSCs were recorded at −70 mV and NMDAR EPSCs were recorded at +40 mV and measured at 100 ms after the stimulus to ensure no contamination from the AMPARs. Deleting CaMKIIα reduced the AMPAR EPSC by approximately 50% (Fig. 1i, l) and the NMDAR EPSC by 30% (Fig. 1m, p). No change in paired pulse ratio, a measure of presynaptic release probability, was seen (Supplementary Fig. 1a, b). Furthermore there was no change in the decay kinetic of the pharmacologically isolated NMDAR EPSC (Supplementary Fig. 1c). The AMPA/NMDA ratio was reduced (Supplementary Fig. 1d), as expected, since the reduction in AMPAR currents was more severe than the reduction in the NMDAR currents. Deletion of CaMKIIβ had no effect on AMPAR EPSCs (Fig. 1j, l) or on NMDAR EPSCs (Fig. 1n, p). Deletion of both CaMKIIα and CaMKIIβ (DKO) resulted in a further decrease in the AMPAR EPSC (Fig. 1k, l), but no further decrease in the NMDAR EPSC (Fig. 1o, p).

What is the relative importance of the two CaMKII subunits in synaptic plasticity? In our hands LTP of evoked EPSCs is unreliable in slice culture. Thus, for these experiments we turned to in utero electroporation in mice at embryonic day 15 (E15),

**Fig. 1** CaMKIIα is required for basal synaptic transmission. **a** Timeline of experiment, map of the CRISPR constructs transfected biolistically and scheme of the electrophysiological approach. The confocal image caption shows an example of a transfected pyramidal neuron. Scale bar 10 μm. **b** Western blot for CaMKII α and β isoforms from dissociated neuronal cultures infected with lentiCRISPR for the respective isoform. Images have been cropped for presentation. Full size images are presented in supplementary Fig. 11. **c−h** Immunofluorescence in dissociated neuronal cultures transfected with CRISPR_CaMKIIα (red channel) and detected with CaMKIIα antibody (green channel) (**c−e**) or transfected with CRISPR_CaMKIIβ (red channel) and detected with CaMKIIβ antibody (green channel) (**f−h**). Scale bar 10 μm. **i−k** Scatterplots show amplitudes of AMPAR EPSC for single pairs (open circles) of control and transfected cells of CRISPR_CaMKIIα (**i**, $n = 32$ pairs), CRISPR_CaMKIIβ (**j**, $n = 17$ pairs) and CRISPR_DKO (**k**, n = 46 pairs). Filled circles indicate mean amplitude ± SEM (**i**, Control = 116.3 ± 9.6; CRISPR_CaMKIIα = 58.03 ± 6.1, $p < 0.0001$. **j**, Control = 106.6 ± 11.6; CRISPR_CaMKIIβ = 109.6 ± 19.66, $p = 0.90$. **k**, Control = 106.6; CRISPR_DKO = 49.3 ± 3.4, $p < 0.0001$). **l** Bar graph of ratios normalized to control (%) summarizing the mean ± SEM of AMPAR EPSCs for values represented in **i** (56.18 ± 5, $p < 0.0001$), **j** (98.23 ± 10, $p = 0.58$) and **k** (41.6 ± 3.2, $p < 0.0001$). **m−o** Scatterplots show amplitudes of NMDAR EPSCs for single pairs (open circles) of control and transfected cells of CRISPR_CaMKIIα (**m**, $n = 29$ pairs), CRISPR_CaMKIIβ (**n**, $n = 17$ pairs) and CRISPR_DKO (**o**, $n = 44$ pairs). Filled circles indicate mean amplitude ± SEM (**m**, Control = 45.5 ± 3.4; CRISPR_CaMKIIα = 28.2 ± 2.2, p = 0.0001. **n**, Control = 47.3 ± 8.5; CRISPR_CaMKIIβ = 45.6 ± 8.7, $p = 0.88$. **o**, Control = 72 ± 4.6; CRISPR_DKO = 44.3 ± 3.2, $p < 0.0001$). **p** Bar graph of ratios normalized to control (%) summarizing the mean ± SEM of NMDAR EPSCs for values represented in **m** (74.8 ± 7, $p = 0.0003$), **n** (109.7 ± 16, $p = 0.75$) and **o** (66 ± 5, $p < 0.0001$). Raw amplitude data from dual cell recordings (scatterplots) were analyzed using Wilcoxon signed rank test (p values indicated above). Normalized data (bar graphs) were analyzed using a one-way ANOVA and $p$ values were calculated with the Mann−Whitney test (***$p$ < 0.0001; *$p < 0.05$). Scale bars: 50 ms, 50 pA. See also Supplementary Fig. 1

making acute slices at P20−28 (Fig. 2a). Loss of CaMKIIα resulted in a substantial decrease in the AMPAR EPSC (Fig. 2b), confirming our slice culture results. LTP, induced by a pairing protocol, was absent (Fig. 2c). It is important to note that, as in a previous study from this lab[27], there was a variable, but systematic, run up of the EPSCs (~50%) either when pairing was carried out in the presence of APV (Supplementary Fig. 2) or when no pairing was performed[27]. Experiments in the presence of

APV were interleaved throughout this study, which were averaged together and this APV-insensitive run up is plotted (light gray circles) in all of the LTP graphs. Although in utero deletion of CaMKIIβ had no effect on basal synaptic transmission (Fig. 2d), confirming our results in slice culture, it did block LTP (Fig. 2e), similar to the deletion of CaMKIIα. In utero deletion of both CaMKIIα and CaMKIIβ recapitulated the large depression in AMPAR EPSCs (Fig. 2f) and LTP was absent (Fig. 2g).

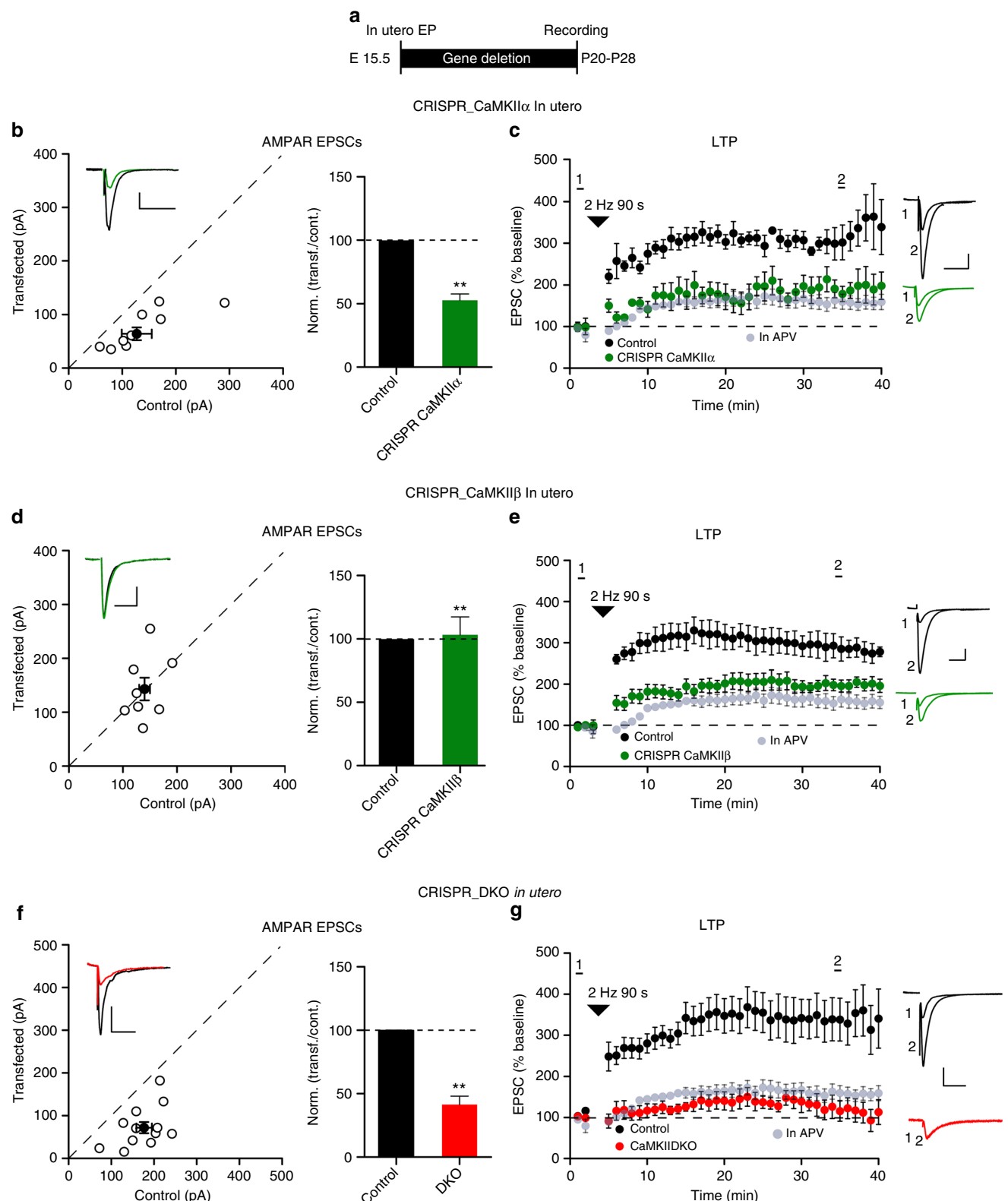

**Fig. 2** Elimination of CaMKII blocks LTP. **a** Experimental timeline. **b** Scatterplot and bar graph of ratios normalized to control of the AMPAR EPSCs for single pairs (open circles) of control and transfected CRISPR_CaMKIIα cells. Filled circle represents mean amplitude ± SEM (Control = 127.3 ± 28.9; CRISPR_CaMKIIα = 67.29 ± 12.7, $n = 8$ pairs, $p = 0.015$). Bar graph of ratios normalized to control (%) summarizing the mean ± SEM of AMPAR EPSCs (51 ± 5, $p < 0.001$). **c** Plots show mean ± SEM AMPAR EPSC amplitude of control (black) and transfected (green) CRISPR_CaMKIIα pyramidal neurons normalized to mean AMPAR EPSC amplitude before LTP induction (arrow) (Control, $n = 6$; CRISPR_CaMKIIα, $n = 7$, $p = 0.029$ at 35 min). **d** Scatterplot and bar graph of ratios normalized to control of the AMPAR EPSCs for single pairs (open circles) of control and transfected CRISPR_CaMKIIβ cells. Filled circle represents mean amplitude ± SEM (Control = 140.4 ± 21.4; CRISPR_CaMKIIβ = 143.5 ± 21.7, $n = 8$ pairs, $p = 0.84$). Bar graph of ratios normalized to control (%) summarizing the mean ± SEM of AMPAR EPSCs (102 ± 15, $p = 0.87$). **e** Plots show mean ± SEM AMPAR EPSC amplitude of control (black) and transfected (green) CRISPR_CaMKIIβ pyramidal neurons normalized to the mean AMPAR EPSC amplitude before LTP induction (arrow) (Control, $n = 7$; CRISPR_CaMKIIβ, $n = 7$, $p = 0.011$ at 35 min). **f** Scatterplot and bar graph of ratios normalized to control of the AMPAR EPSCs for single pairs (open circles) of control and transfected CRISPR_DKO cells. Filled circle represents mean amplitude ± SEM (Control = 81.9 ± 9.4; DKO = 39.1 ± 6.43, $n = 11$ pairs, $p = 0.001$). Bar graph of ratios normalized to control (%) summarizing the mean ± SEM of AMPAR EPSCs (102 ± 15, $p < 0.001$). **g** Plots show mean ± SEM AMPAR EPSC amplitude of control (black) and transfected (red) CRISPR_DKO pyramidal neurons normalized to the mean AMPAR EPSC amplitude before LTP induction (arrow) (Control, $n = 13$; CRISPR_DKO, $n = 8$, $p = 0.0084$ at 35 min). Gray plots represent mean ± SEM AMPAR EPSC amplitude of LTP induction in APV 50 μM. Sample AMPAR EPSC current traces from control (black) and electroporated neurons (green or red) before and after LTP are shown to the right of each graph. Raw amplitude data from dual cell recordings were analyzed using Wilcoxon signed rank test ($p$ values indicated above). Normalized data were analyzed using a one-way ANOVA followed by a Mann−Whitney test to calculate $p$ values (**$p < 0.001$). Mann−Whitney test was used also to compare LTP at 35 min ($p$ values indicated above). Scale bars: 50 ms, 50 pA. See also Supplementary Fig. 3

To determine the relative importance of CaMKIIα and CaMKIIβ in baseline transmission and LTP, we attempted to rescue the defects caused by the DKO by expressing back CaMKII subunits (Fig. 3a, b). CaMKIIα fully rescued both the AMPAR (Fig. 3c, e) and NMDAR EPSCs (Fig. 3f, h). To examine LTP we used in utero electroporation (Fig. 3i). CaMKIIα fully rescued LTP (Fig. 3j) in the DKO, indicating that CaMKIIβ is not required for the full expression of LTP. In contrast to CaMKIIα, the depression in synaptic transmission caused by the DKO was only partially rescued by expression of CaMKIIβ (Fig. 3d, e, g, h). Furthermore, CaMKIIβ was unable to rescue LTP, either in the DKO (Fig. 3k) or in the CaMKIIα deletion (Supplementary Fig. 3b). These findings with CaMKIIβ are consistent with the lower expression levels of this isoform under normal conditions, and further suggest that it is not as effective as CaMKIIα in maintaining basal synaptic transmission or LTP. One concern is the presence of multiple splice variants of CaMKIIβ[28] all of which are deleted by CRISPR. Perhaps the failure to rescue the defects observed with deleting CaMKIIβ is that the splice variant we use[29] is unable to substitute for all of the other slice variants. To test for this scenario, we deleted CaMKIIβ and then asked whether expression of our CaMKIIβ construct could fully rescue LTP. Indeed, there was a full rescue of LTP (Supplementary Fig. 3c).

Are the effects of deleting CaMKII restricted to an early developmental stage when synaptogenesis is high, or is CaMKII required for maintaining synaptic function at mature synapses? To address this we repeated the DKO experiments in the adult hippocampus by expressing the gRNA with lentivirus in CA1 pyramidal cells (Fig. 4a, b) in the Cas9 knockin mouse[30]. Simultaneous paired recordings were made from a transfected cell and a neighboring control cell (Fig. 4c). We found the same depression in both AMPAR (Fig. 4d, e) and NMDAR synaptic responses (Fig. 4f, g). LTP was also abolished in these cells (Fig. 4h). We also analyzed the effect of lentivirus-mediated expression of CaMKII DKO gRNA in dentate gyrus (Fig. 4i). The results, both in terms of basal AMPAR transmission (Fig. 4j, k) and LTP (Fig. 4l), were identical to those observed in the CA1 region. These findings indicate that CaMKII is required to maintain synaptic transmission at stable adult synapses.

**The role of kinase activity in the actions of CaMKIIα.** Many studies have suggested that, in addition to its role as a kinase, CaMKII also has a structural/scaffolding component. To distinguish between these two components we first examined the role of CaMKIIα by expressing two different mutated CaMKIIα

constructs in the DKO background (Fig. 5a, b). Expression of the Thr286 autophosphorylation null CaMKIIα T286A[13,31] or kinase dead CaMKIIα K42R[32] mutants failed to rescue the AMPAR defect (Fig. 5c−e), but fully rescued the NMDAR defect (Fig. 5f−h). These results indicate that CaMKIIα basal kinase activity is required to maintain AMPAR-dependent synaptic transmission. On the other hand, CaMKIIα-dependent maintenance of the NMDAR EPSC is entirely independent of kinase activity, suggesting a structural/scaffolding role. It is important to note that the inability to rescue the AMPAR EPSC is not due to the lack of expression of the CaMKIIα mutants, because there was a full rescue of the NMDAR EPSC. Immunofluorescence experiments in dissociated neuronal cultures confirm the decrease in surface GluN2B containing receptors in the DKO background (Supplementary Fig. 4a−c). This phenotype is rescued by both WT CaMKIIα and mutants (CaMKIIα T286A and CaMKIIα K42R).

Given the profound effects of CaMKIIα and CaMKIIβ deletion on excitatory synaptic function, we used super-resolution microscopy to examine the morphology of dendritic spines (Supplementary Fig. 4d), which receive the large majority of excitatory synapses (Supplementary Fig. 4e). Although there was no change in the density of spines (Supplementary Fig. 4e, f), we found a substantial reduction in head diameter (Supplementary Fig. 4e, g) and an increase in the length of the spine neck (Supplementary Fig. 4e, h). Expression of CaMKIIα K42R rescued the DKO effect on spine neck, but not on the head diameter (Supplementary Fig. 4e−h), indicating that kinase activity is required for maintaining spine head diameter. Given the correlation between spine size and number of AMPARs[33−35], one might expect a decrease in the size of miniature EPSCs (mEPSCs) in cells lacking CaMKII. Indeed, we recorded a significant reduction in the amplitude of mEPSCs, but no change in frequency (Supplementary Fig. 1e−l), consistent with the conserved spine density.

To further elucidate the role of CaMKII in spine morphology, we analyzed structural LTP (sLTP) using two-photon glutamate uncaging in organotypic slices biolistically transfected with GFP, gRNAs, and Cas9 (and CaMKIIα T286A for replacement). Employing an established sLTP protocol consisting of 30 pulses of glutamate uncaging at 0.5 Hz, in the absence of $Mg^{2+}$[36,37], we analyzed changes in spine volume in the WT, CaMKII DKO, and CaMKIIα T286A replacement. We confirmed (Fig. 5i−k) that sLTP consists of a fast transient phase with a large increase in spine volume, which lasts for approximately 8 min, followed by a long-term 1.5-fold enhancement phase[37]. We repeated these experiments in CaMKII DKO transfected neurons and found

that the transient phase remained intact, while the stable enhancement was abolished (Fig. 5i, j and Supplementary Fig. 5d). The same result was obtained in CaMKIIα KO (Supplemental Fig. 5f, h, i, k) and CaMKIIβ KO neurons (Supplementary Fig. 5g, h, j, k), highlighting the importance of CaMKIIβ in sLTP. These results strongly confirm our data obtained in electrophysiology experiments (Fig. 2e). What, then, is required for sLTP? Application of NMDAR inhibitors CPP or APV, before uncaging glutamate abolished spine volume changes as expected (Supplementary Fig. 6a, c, f). To determine if $Ca^{2+}$ is

required, we repeated the experiment in WT GFP neurons in a solution that prevented $Ca^{2+}$ influx (nominally 0 $Ca^{2+}$ with EGTA 10 mM). This treatment completely prevents sLTP (Supplementary Fig. 6d, f). To confirm that this process is strictly dependent on $Ca^{2+}$, we switched the solution to one containing $Ca^{2+}$ (4 mM) and measured sLTP from neighbor spines. sLTP was fully rescued by $Ca^{2+}$ (Supplementary Fig. 6e, f). Finally we induced sLTP in neurons expressing CaMKIIα T286A GFP : the transient phase is identical to WT, but long-term sLTP is not maintained (Fig. 5i−k and Supplementary

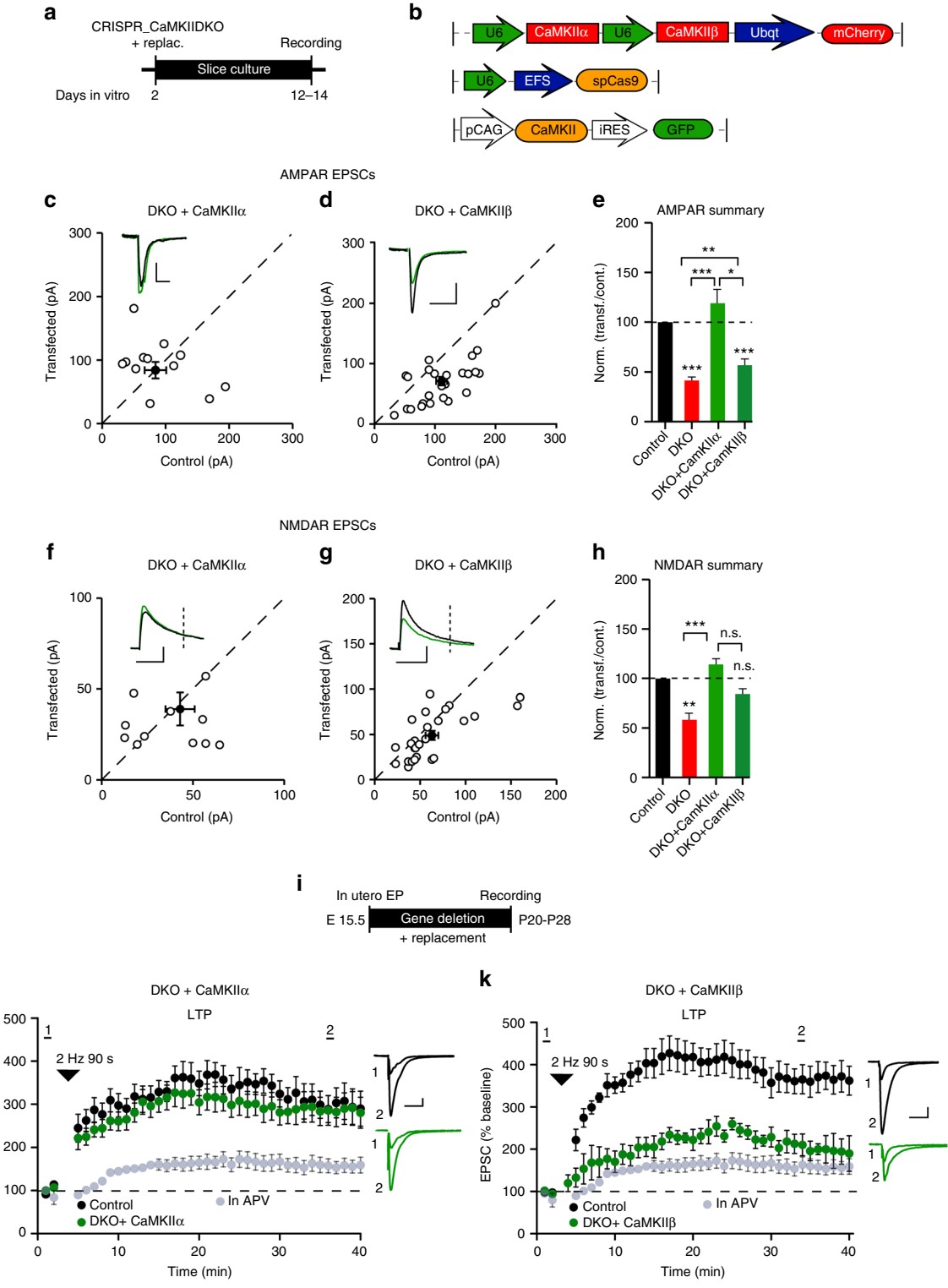

**Fig. 3** Molecular replacement confirms the key role of CaMKIIα. **a** Timeline of the experiment. **b** Representative maps of constructs. **c, d** Scatterplots show amplitudes of AMPAR EPSCs for single pairs (open circles) of control and transfected cells of DKO + CaMKIIα (**c**, n = 11 pairs) and DKO + CaMKIIβ (**d**, n = 26 pairs). Filled circles indicate mean amplitude ± SEM (**c**, Control = 90.8 ± 16.8; DKO + CaMKIIα = 88.6 ± 13.2, p = 0.96; **d**, Control = 111.1 ± 9; DKO + CaMKIIβ = 71.2 ± 7.9, p = 0.002). **e** Bar graph of ratios normalized to control (%) summarizing the mean ± SEM of AMPAR EPSCs of values represented in **c** (132 ± 30, p = 0.76) and **d** (66 ± 6, p = 0.0009). DKO data (red bar) from Fig. 1l are included in the graph. **f−h** Scatterplots show amplitudes of NMDAR EPSCs for single pairs (open circles) of control and transfected cells of DKO + CaMKIIα (**f**, n = 11 pairs) and DKO + CaMKIIβ (**g**, n = 26 pairs). Filled circles indicate mean amplitude ± SEM (**f**, Control = 43.5 ± 8.4; DKO + CaMKIIα = 38.7 ± 9.5, p = 0.51; **g**, Control = 63.4 ± 7; DKO + CaMKIIβ = 49.5 ± 5, p = 0.078). **h** Bar graph of ratios normalized to control (%) summarizing the mean ± SEM of NMDAR EPSCs of values represented in **f** (114 ± 24, p = 0.76) and **g** (82 ± 7, p = 0.09). DKO data (red bar) from Fig. 1p are included in the graph. **i** Timeline of the in utero electroporation experiment. **j** Plots show mean ± SEM AMPAR EPSC amplitude of control (black) and transfected (green) DKO + CaMKIIα pyramidal neurons normalized to the mean AMPAR EPSC amplitude before LTP induction (arrow) (Control, n = 5; DKO + CaMKIIα, n = 8, p = 0.289 at 35 min). **k** Plots show mean ± SEM AMPAR EPSC amplitude of control (black) and transfected (green) DKO + CaMKIIβ pyramidal neurons normalized to the mean AMPAR EPSC amplitude before LTP induction (arrow) (Control, n = 5; DKO + CaMKIIβ, n = 5, p = 0.017 at 35 min). Gray plots represent mean ± SEM AMPAR EPSC amplitude of LTP induction in APV 50 μM. Raw amplitude data from dual cell recordings were analyzed using Wilcoxon signed rank test (p values above). Normalized data were analyzed using a one-way ANOVA and p values calculated with the Mann−Whitney test (***p < 0.0001; **p < 0.001; *p < 0.05). Mann–Whitney test was used also to compare LTP at 35 min (p values indicated above). Scale bars: 50 ms, 50 pA. See also Supplementary Fig. 3

Fig. 5e). These results indicate that CaMKIIα activity is necessary for long-term sLTP, but is not for transient sLTP.

The results in Fig. 2 indicate that deleting CaMKII blocks LTP. However, it is important to note that the NMDAR EPSC is also reduced, raising the possibility that the blockade in LTP is secondary to the effect on NMDARs. To address this issue we took advantage of the CaMKIIα mutations (T286A and K42R), which fully rescue the NMDAR current (Fig. 5f−h). Thus we carried out an additional series of in utero electroporation experiments using the DKO strategy (Fig. 6a, b). Similar to the slice culture results, both the AMPAR (Fig. 6g) and NMDAR EPSCs (Fig. 6c, e) were substantially reduced. We then examined the ability of CaMKIIα mutants, T286A and K42R, to rescue basal transmission and LTP. In agreement with our results in slice culture CaMKIIα T286A failed to rescue the AMPAR EPSCs (Fig. 6f, g), but it fully rescued the NMDAR EPSC (Fig. 6d, e). In cells expressing the T286A mutant LTP was blocked (Fig. 6h). This finding indicates that the reduction in the NMDAR function in the CaMKII deletions cannot account for the blockade of LTP (Fig. 2) and that CaMKIIα kinase activity is, indeed, essential for LTP. In utero electroporation of CaMKIIα K42R had the same effects as CaMKIIα T286A on baseline synaptic transmission (Supplementary Fig. 7b−e), but the effects on LTP (Supplementary Fig. 7f) were not as dramatic as the effects of the CaMKIIα T286A mutant.

**The synaptic actions of CaMKII require binding to NMDARs.** Considerable evidence has demonstrated that CaMKII binds to NMDARs[38−40] and accumulates at postsynaptic sites[40,41]. Residues within the GluN2B C-terminal tail that are critical for this interaction have been mapped, and either overexpression[42] or knockin of mutants[43] disrupting CaMKII binding impairs LTP. Two adjacent sites on the kinase domain of CaMKII, termed the S-site and T-site, are required for the binding to NR2B. An initial weak S-site occupation is followed by a more stable interaction with the T-site[40]. We adopted two different critical mutations in CaMKIIα to prevent NR2B binding: F98K (S-site) and I205K (T-site). The physiological consequences of mutating these two critical sites on CaMKIIα[40,44] (Fig. 7b) remain unknown. We expressed the F98K and I205K mutants on the DKO background in slice culture (Fig. 7a, b). Remarkably, we find that, although the kinase activity of these mutants is normal[40,44], their effects were identical to CaMKIIα T286 and CaMKIIα K42R mutants, failing to rescue AMPAR EPSCs (Fig. 7c, d, f), but fully rescuing NMDAR EPSCs (Fig. 7g, h, j). Combining the CaMKIIα I205K mutation with the kinase dead mutation (K42R) yielded result indistinguishable from the I205K mutation on its own (Fig. 7e, f, i, j). These results suggest that CaMKIIα binding to the NMDAR is essential for CaMKII to affect synaptic transmission. In

contrast to the negative results with the CaMKIIα F98K and I205K mutants, expression of WT CaMKIIα fully rescued the defects caused by deleting endogenous CaMKII (Fig. 7f, j). We next examined the effect of disrupting the binding of CaMKIIα to NMDARs on LTP. We used in utero electroporation (Fig. 8a) of the CaMKIIα I205K mutant (Fig. 8b) on the DKO background and recorded from acute slices at P20-28. Similar to the results in slice culture, the I205K mutant failed to rescue baseline AMPAR transmission (Fig. 8c, d), but rescued NMDAR transmission (Fig. 8e, f). Preventing CaMKII binding to NMDARs abolished LTP (Fig. 8g).

While the CaMKIIα F98K and I205K mutations block the binding of CaMKII to the GluN2B receptor[40,44], they also affect binding to itself[45] as well as other proteins such as densin[46] and GluN1[47]. These others binding deficits could explain the more dramatic effects we observe on LTP compared to previous results in which mutations were made in the GluN2B subunits[42,43]. To compare more directly our results to earlier studies, we carried out a series of molecular replacement experiments with the GluN2B subunit (Fig. 9a, b). Deleting GluN2B causes a dramatic reduction in the NMDAR EPSCs (Fig. 9g, l) and an enhancement of AMPAR EPSCs (Fig. 9c, k), as expected from previous results[48]. We next attempted to rescue the changes by expressing wild-type GluN2B. In this case the NMDAR current was fully rescued (Fig. 9h, l) and there was no change in the AMPAR current (Fig. 9d, k). The expression of the mutant form of GluN2B that abolishes the binding of CaMKIIα (Fig. 9a, b)[43] fully rescued the NMDAR EPSC (Fig. 9i, l), demonstrating that this mutation had no effect on the function of the NMDAR. Importantly, the AMPAR EPSCs are reduced (Fig. 9e, k) to an extent like that seen with the Thr286 autophosphorylation null mutant of CaMKIIα (T286A) (Fig. 9k). This finding indicates that the binding of CaMKIIα to the NMDAR is necessary for its role in maintaining basal synaptic transmission. Finally, and most importantly, the dramatic enhancement of synaptic transmission observed with the kinase active CaMKIIα T305A/T306A muta-tion (Fig. 9k) (see below) is entirely prevented (Fig. 9f, k). NMDAR EPSCs in the CaMKIIα T305A/T306A replacement were not different from control (Fig. 9j). Together these findings establish that for CaMKII to have any effect on synaptic function, except for its effects on NMDARs, it must be docked at the PSD by its binding to the NMDAR.

**CaMKII constitutive activity and the maintenance of LTP.** One of the most controversial aspects of CaMKII function is whether it can maintain LTP following the transient rise in calcium during the induction of LTP. It has previously been shown that autop-hosphorylation of CaMKIIα on T305 and T306 in the $Ca^{2+}$

/CaM binding domain (Supplementary Fig. 8b) prevents activation by $Ca^{2+}$/CaM[49,50], whereas preventing phosphorylation of these residues greatly enhances $Ca^{2+}$/CaM-driven CaMKII activity[51,52]. We found that CaMKIIα, in which either T305, or T306, or both were mutated to aspartate, thus mimicking phosphorylation, all failed to rescue AMPAR currents in the DKO (Supplementary Fig. 8c−f), although they all fully rescued the NMDAR currents (Supplementary Fig. 8g−j). These results confirm that these two amino acids are critical for the $Ca^{2+}$/CaM activation of CaMKIIα.

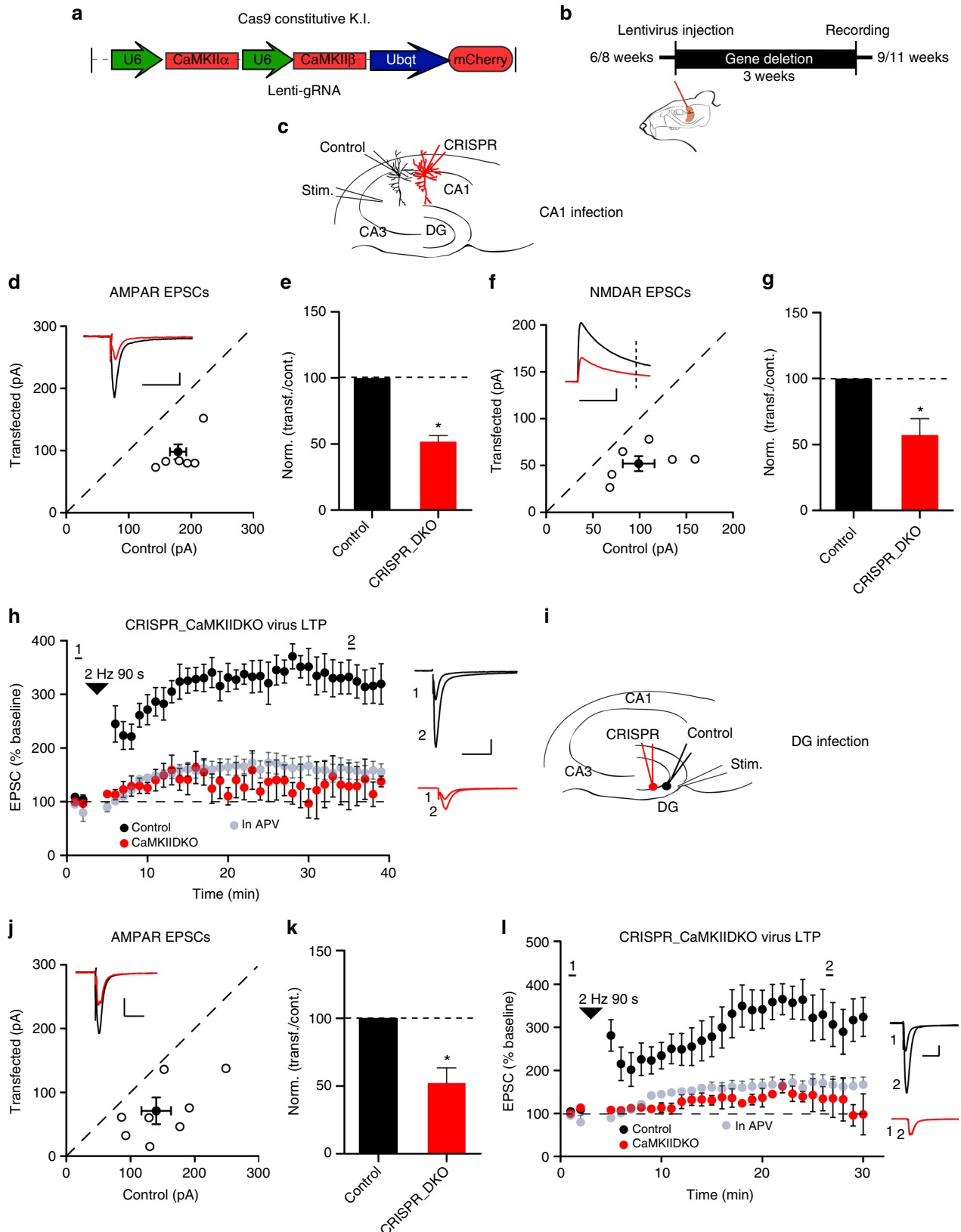

**Fig. 4** Basal synaptic transmission and LTP in adult mice hippocampus require CaMKII. **a** Map of the CRISPR construct used to transduce the lentivirus and mouse model used in the experiment. **b** Timeline of the experiment. **c** Scheme of the dual cell recording experiment approach. **d** Scatterplots show amplitudes of AMPAR EPSCs for single pairs (open circles) of control and transfected cells of DKO ($n = 6$ pairs). Filled circle indicates mean ± SEM (**d**, Control = 179.3 ± 13.3; DKO = 93.8 ± 14.6, $p < 0.001$). **e** Bar graph of ratios normalized to control (%) summarizing the mean ± SEM of AMPAR EPSCs of values represented in **d** (51 ± 4.7, $p = 0.02$). **f** Scatterplots show amplitudes of NMDAR EPSCs for single pairs (open circles) of control and transfected cells of DKO ($n = 6$ pairs). Filled circle indicates mean ± SEM (**f**, Control = 98 ± 17; DKO = 52.8 ± 9, $p < 0.05$). **g** Bar graph normalized to control (%) summarizing the mean ± SEM of NMDAR EPSCs of values represented in **f** (62.9 ± 8.5, $p = 0.04$). **h** (Control, $n = 6$; DKO, $n = 4$, $p = 0.01$ at 35 min). Scale bars: 50 pA, 50 ms. **i** Scheme of the dual cell recording experiment approach, in dentate gyrus. **j** Scatterplot showing amplitudes of AMPAR EPSCs for single pairs (open circles) of control and transfected cells of DKO ($n = 8$ pairs). Filled circle indicates mean ± SEM. **j** Control = 140.9 ± 24; DKO = 75.1 ± 21, $p < 0.001$. **k** Bar graph of ratios normalized to control (%) summarizing the mean ± SEM of AMPAR EPSCs of values represented in **j** (52.9 ± 11, $p = 0.03$). **l** Plots showing mean ± SEM AMPAR EPSC amplitude of control (black) and transfected CRISPR_DKO (red) pyramidal neurons normalized to the mean AMPAR EPSC amplitude before LTP induction (arrow) (Control, $n = 6$; DKO, $n = 4$, $p = 0.01$ at 30 min). Gray plots represent mean ± SEM AMPAR EPSC amplitude of LTP induction in APV 50 μM. Raw amplitude data from dual cell recordings were analyzed using Wilcoxon signed rank test ($p$ values indicated above). Normalized data were analyzed using a one-way ANOVA and $p$ values calculated with the Mann−Whitney test (*$p < 0.01$). Mann−Whitney test was also used to compare LTP at 35 min ($p$ values indicated above). Scale bars: 50 ms, 50 pA

We next examined the phospho null mutants (T305A/T306A), by overexpressing them on a WT background (2–3 days). We compared the effects of WT CaMKIIα, CaMKIIα T286D-T305A/T306A, and CaMKIIα T305A/T306A in slice culture (Fig. 10a, b). Overexpressing WT CaMKIIα had no effect on synaptic transmission (Fig. 10c, f), whereas overexpressing CaMKIIα T305A/T306A caused a three-fold enhancement (Fig. 10d, f). The constitutively active CaMKIIα T286D-T305A/T306A (CA CaM-KIIα) mutant[50] caused an identical enhancement (Fig. 10e, f). However, the mechanisms by which these two constructs enhance synaptic transmission are fundamentally different. As expected the activity of CaMKIIα T286D-T305A/T306A is independent of $Ca^{2+}$/CaM and NMDAR activity (Fig. 10h, k). On the other hand, the enhancement observed with CaMKIIα T305A/T306A is entirely dependent on ongoing NMDAR activity, since incubation of slices with APV prevented the enhancement (Fig. 10g, k). This finding is consistent with the increased affinity of this mutation to $Ca^{2+}$/CaM. As a result cells expressing this mutant are more sensitive than control neurons to the ongoing spontaneous activity and NMDAR-dependent plasticity in slice cultures[50]. Preventing CaMKIIα from binding to NMDARs with the CaMKIIα I205K mutation completely abrogates its ability to enhance synaptic transmission (Fig. 10i, k). Even more remarkable is the finding that the enhancement caused by constitutively active CaMKIIα T286D-T305A/T306A is also completely prevented by the I205K mutation suggesting that the binding of CaMKII to NMDARs is essential for the actions of CaMKII (Fig. 10j, k). This finding highlights the fact that the presence of a constitutively active kinase in the spine is not sufficient to enhance synaptic function—it requires the precise targeting provided by binding to NMDARs.

The findings also indicate that CaMKIIα T305A/T306A is considerably more sensitive to $Ca^{2+}$/CaM than WT CaMKIIα and that its actions require NMDAR activation. Is NMDAR activity only required initially, as would be predicted by the model in which CaMKIIα activity becomes independent of $Ca^{2+}$? In this case, one would predict that the acute application of APV would have little or no effect on the enhancement, once the enhancement is initiated and locked in. We first examined the acute application of APV on the enhancement caused by expression of the constitutive active CaMKIIα T286D-T305A/T306A. Neither short term (10 min) nor long-term (2 h) application of APV had any effect on the enhancement (Fig. 10l). In striking contrast, APV completely reversed the enhancement caused by CaMKIIα T305A/T306A over a 2-hour period (Fig. 10m). Interestingly, voltage-clamping cells at −70 mV expressing CaMKIIα T305A/T306A in the absence of APV was sufficient to reverse the enhancement over a 90-min period

(Supplementary Fig. 9). This is to be expected, since clamping the cells prevents NMDAR activation. Similar results were obtained when these experiments were carried out on the DKO background. Thus the enhancing action of CaMKIIα T305A/T306A (Supplementary Fig. 10c, f) was absent in the presence of APV for 10DIV (Supplementary Fig. 10d, f) or for a short period (2 h) (Supplementary Fig. 10e, f). An example of the time course of APV's action is shown in Supplementary Fig. 10g. As would be expected the CaMKIIα T305A/T306A construct fully rescued NMDA EPSCs on the DKO background (Supplementary Fig. 10h, i). Thus our experiments failed to uncover a constitutive $Ca^{2+}$/CaM-independent component to the action of CaMKII.

## Discussion

Despite the numerous studies that have been carried out on the properties of CaMKII over that past two decades, much confusion remains. In this study we took advantage of CRISPR technology to systematically and rigorously probe the role of CaMKII in synaptic function. CRISPR provides a way to relatively rapidly and completely delete synaptic proteins[53]. Coupled with the use of slice culture and in utero electroporation, along with simultaneous paired recordings, we were able to resolve a number of the outstanding issues in this field. In brief, CaMKIIα, but not CaMKIIβ, is required to maintain basal AMPAR and NMDAR synaptic transmission. AMPAR transmission requires kinase activity, but NMDAR transmission does not, indicating a scaffolding role for the CaMKII protein instead. LTP was abolished in the absence of CaMKIIα and β, in the DKO, and with the molecular replacement of the Thr296 - autophosphorylation null mutant (T286A). All aspects of CaMKIIα signaling examined in this study, except for the maintenance of NMDAR synaptic currents, require binding to the NMDAR, emphasizing the essential role of this receptor as a synaptic signaling hub. Finally, evidence is presented that the maintenance of the CaMKII-induced synaptic enhancement does not involve $Ca^{2+}$-independent kinase activity of CaMKII.

A number of the findings reported in this study differ quantitatively, and in some cases qualitatively, from many previous studies, especially those using genetic approaches. For instance, we find that deleting CaMKIIα fully blocks LTP and that disrupting the interaction of CaMKII with the NMDAR entirely prevents the synaptic actions of CaMKII. In addition, previous genetic studies failed to observe changes in baseline transmission of either the AMPAR or NMDAR EPSCs. One possibility is compensation with the germline KOs. However, it is unclear how this could occur. It seems most unlikely that CaMKIIβ could compensate for the absence of CaMKIIα, because it is unable to support LTP. The only clear difference between our study and

most previous studies is that we used single cell manipulations, whereas most previous studies used global genetic manipulations. There are a couple studies, one on neuroligin[54] and the other on ephrin-B3[55] that directly compared these two conditions. They found that global manipulations of these synaptic adhesion molecules had no effect on synapse number, whereas single cell manipulations did. They propose that there may be a transcellular competitive process. However, it is unclear how such a mechanism would work in the present study.

If CaMKII serves as a memory molecule by maintaining enhanced synaptic transmission, one would expect that its deletion would result in a deficit in synaptic transmission. Surprisingly, CaMKII knockouts[12,26] or CaMKII T286A knockin mice[13,14] display no change in baseline transmission, despite LTP inhibition. Although early studies found little effect of classical CaMKII inhibitors including peptides and small molecules on baseline transmission[9,10,22], more recent studies with new peptide inhibitors have reported a depression in basal transmission[23,24],

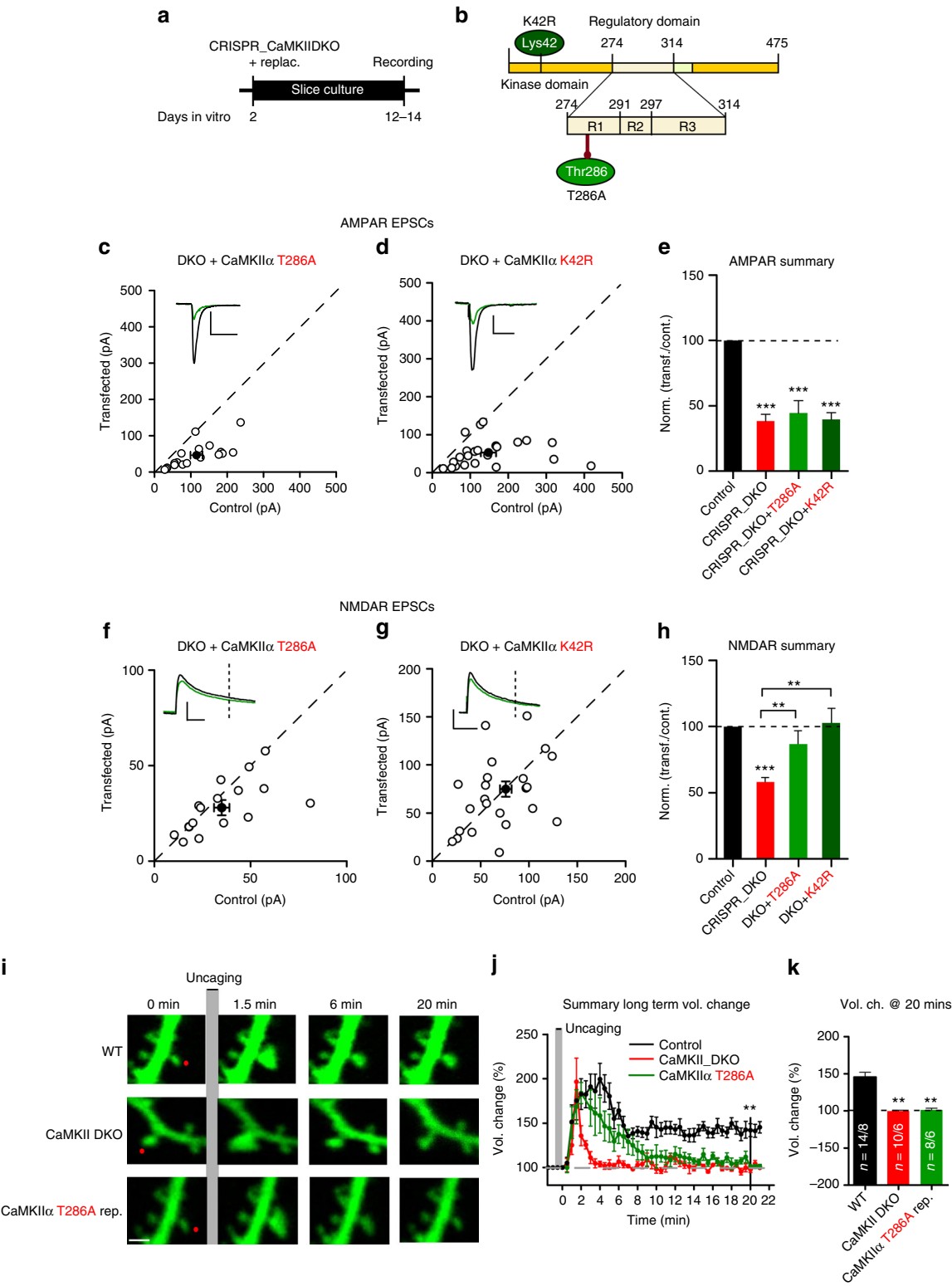

**Fig. 5** CaMKIIα T286A and K42R mutated forms rescue NMDAR, but not AMPAR, EPSCs or sLTP. **a** Timeline of the experiment. **b** Scheme of the structural organization of CaMKIIα showing point mutations (K42R dark green and T286A light green). **c**, **d** Scatterplots showing amplitudes of AMPAR EPSCs for single pairs (open circles) of control and transfected cells of DKO + CaMKIIα T286A (**c**, n = 18 pairs), and DKO + CaMKIIα K42R (**d**, n = 23 pairs). Filled circle indicate mean ± SEM. (**c**, Control = 110.9 ± 17.8; DKO + CaMKIIα T286A = 49.7 ± 9.1 $p < 0.0001$; **d**, Control = 148.4 ± 20.4; DKO + CaMKIIα K42R = 53.5 ± 7.5 $p < 0.0001$). **e** Bar graph of ratios normalized to control (%) summarizing the mean ± SEM of AMPAR EPSCs of values represented in **c** (43.9 ± 6, $p = 0.0009$) and **d** (45.4 ± 5.1, $p < 0.0001$). DKO data (red bar) from Fig. 1l are included in the graph. **f**, **g** Scatterplots showing amplitudes of NMDAR EPSCs for single pairs (open circles) of control and transfected cells of DKO + CaMKIIα K42R (**f**, n = 23 pairs), and DKO + CaMKIIα T286A (**g**, n = 21 pairs). Filled circles indicate mean ± SEM. (**f**, Control = 76.1 ± 6.6; DKO + CaMKIIα K42R = 75.3 ± 8, $p = 0.93$; **g**, Control = 38.6 ± 3.8; DKO + CaMKIIα T286A = 32.6 ± 3, $p = 0.078$). **h** Bar graph of ratios normalized to control (%) summarizing the mean ± SEM of NMDAR EPSCs of values represented in **f** (107 ± 13, $p = 0.84$) and **g** (86.8 ± 6.9, $p = 0.70$). DKO data (red bar) from Fig. 1p are included in the graph. **i** Fluorescence GFP image samples of spine structural plasticity during sLTP. The red dot indicates the spot of glutamate uncaging. Scale bar 1 μm. **j** Long-term spine volume change of WT (black), CaMKII DKO (red) and DKO + CaMKIIα T286A (green). Each point represents the mean ± SEM % of volume change every 30 s. **k** Bar graph of averaged volume change at 20 min. Values represent mean ± SEM as % of baseline volume (Control = 145.5 ± 6; DKO = 100 ± 7.1; DKO + CaMKIIα T286A = 100.1 ± 2.3). Number of samples (spines/neurons) is 14/8 for control cells; 10/6 for DKO, and 8/6 for T286A replacement. Raw amplitude data from dual cell recordings were analyzed using Wilcoxon signed rank test ($p$ values indicated above). Normalized data (including sLTP) were analyzed using a one-way ANOVA followed by the Mann−Whitney test (***$p < 0.0001$; **$p < 0.001$). Scale bars: 50 ms, 50 pA. See also Supplementary Fig. 4−6

but nonspecific effects, especially with bath application, contribute substantially to the depression[25].

In our hands, deletion of CaMKIIα alone or in combination with CaMKIIβ (DKO) with CRISPR/Cas9 caused a dramatic reduction (~50%) in AMPAR EPSCs together with a modest reduction (~30%) in NMDAR EPSCs using both slice culture preparations or following in utero electroporation. A previous study presented evidence that the subunit composition of NMDARs was altered in cells expressing CaMKIIα T286A[56]. However, in our experiments we found no change in the decay kinetics of the NMDAR EPSC in either CaMKIIα KO or in the DKO. Deletion of CaMKIIβ had no effect on basal synaptic transmission, in contrast to a modest reduction reported in a previous study using shRNA[19]. This lack of effect is likely due to the lower expression level of the protein, as well as its reduced activity. However both electrophysiological LTP and two photon sLTP after CRISPR_CaMKIIβ deletion, confirmed a fundamental role of this subunit in LTP (Fig. 2 and Supplementary Fig. 5). What might explain this seemingly contradictory result? As described before, there is an excess of CaMKIIα compared to CaMKIIβ[5] at normal physiological levels. One possible explanation is that, when we delete the β isoform the total amount of CaMKII is substantially reduced such that the residual amount of CaMKIIα is not sufficient for inducing LTP. Nevertheless, the reduced levels of CaMKII would be sufficient to maintain basal transmission. Only with more severe reduction in total CaMKII levels do we see a reduction in basal transmission (see CaMKIIα and DKO deletions (Fig. 1i and k)). Furthermore, deletion of CaMKII in adult hippocampus caused a similar reduction. Thus it is safe to conclude that the strength of excitatory synaptic transmission is dependent on the presence of CaMKII throughout life.

The unusually high amount of CaMKII in the brain and at synapses raised early on the possibility that it might serve a structural role. Indeed, recent evidence has emerged for kinase-independent structural roles for CaMKIIβ[3]. Both the morphological effects of deleting CaMKIIβ[20] and the impairment of CaMKIIα targeting to the PSD in the CaMKIIβ KO mouse can be rescued by a kinase dead mutant of CaMKIIβ[21]. In the present study CaMKIIα T286A and K42R failed to rescue AMPAR basal synaptic transmission or LTP. However, they fully rescued NMDAR basal synaptic transmission, indicating a kinase-independent, structural/scaffolding role for CaMKII. Curiously, unlike all of the other actions of CaMKII, which required binding of CaMKII to the NMDAR, the rescue of the NMDAR current did not.

It is often stated in reviews that CaMKII is both necessary and sufficient for LTP. However, the literature on the necessity of CaMKIIα for LTP does not support such a strong conclusion.

While pharmacological[9–11] and genetic data[12–14,51] show that inhibiting/deleting CaMKII causes a dramatic reduction in LTP, in most studies significant potentiation remains, especially in the genetic models. This raises the possibility of a CaMKII-independent component to NMDAR-dependent LTP. We found that deletion of CaMKIIα alone or together with CaMKIIβ abolished LTP. However, in both of these manipulations, there is a clear reduction in the NMDAR EPSC, raising the possibility that some of the LTP deficit could result from this loss. To circumvent this issue we took advantage of our finding that CaMKIIα T286A and K42R fully rescue the NMDAR defect in the DKO. The T286A mutation mimicked the effect of CaMKIIα deletion, indicating that this subunit fully accounts for NMDAR-dependent LTP. Recently it has been proposed that CaMKIIα T286A retains some kinase activity and that cells expressing T286A can show LTP when the stimulus frequency is increased beyond that typically use for pairing-induced LTP (<2 Hz)[57]. Much to our surprise, substantial LTP remained with the K42R mutation, even though this mutant's effect on baseline transmission was identical to that of T286A. Taken together our findings indicate that CaMKIIα can fully account for NMDAR-dependent LTP. There is thus no need to postulate a CaMKII-independent component.

In addition to the DKO having dramatic effects on baseline synaptic transmission, deleting both isoforms also caused clear changes in dendritic spine morphology. Although the density of spines was unaltered, the size of the spine head was reduced and the spine neck became longer. The reduction in spine head size is consistent with the hypothesis that AMPAR number is directly related to spine size[37] and the reduction in mEPSC amplitude.

In addition to its role in spine morphology, CaMKII is also critical for sLTP. Uncaging glutamate caused a rapid increase in spine volume that relaxed over a 6−8 min period to a stable enhancement, similar to previous studies[37]. In DKO neurons or neurons lacking either CaMKIIα or β the size of the transient enhancement was unaltered, but the decay back to baseline was more rapid and returned to baseline in ~4 min. Neurons expressing the CaMKIIα T286A mutant on DKO background decayed back to baseline more slowly (8–10 min), similar to the kinetics of the transient phase in WT neurons. These deletion studies confirm the fundamental role of CaMKII in sLTP[11,36,37,58]. However, the early transient enhancement was independent of CaMKII, but entirely dependent on NMDAR activation and on $Ca^{2+}$. These finding suggest that $Ca^{2+}$ can directly increase spine volume, presumably by directly engaging the actin cytoskeleton[59,60]. However, the physiological relevance of this CaMKII-independent spine enlargement remains unclear, since pairing induced LTP does not induce a rapid transient

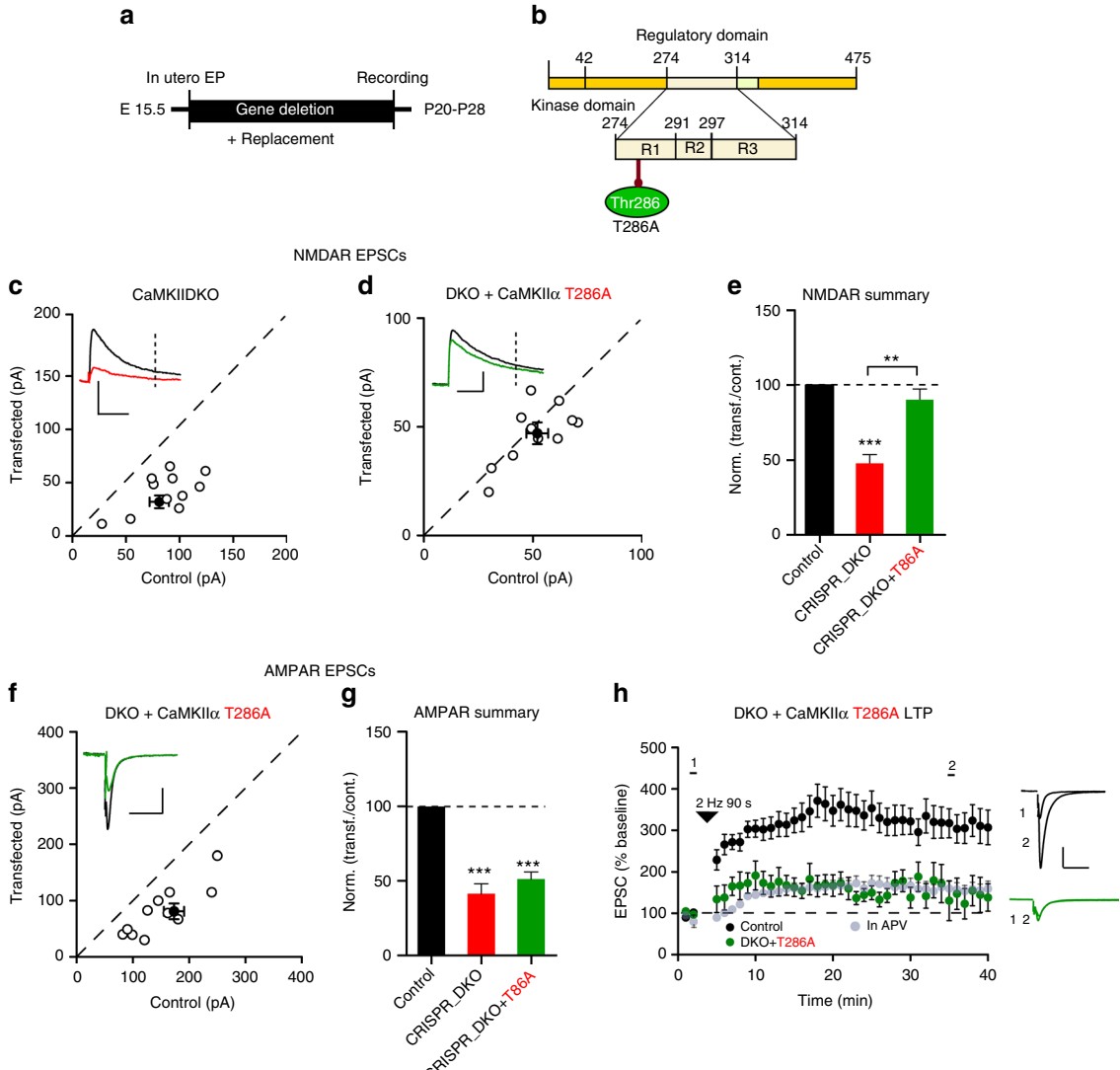

**Fig. 6** Cells expressing CaMKIIα T286A lack LTP. **a** Timeline of the experiment. **b** Scheme of the structural organization of CaMKIIα with the point mutation that makes CaMKII autophosphorylation impaired (T286A light green) indicated. **c**, **d** Scatterplots show single pairs (open circles) of control and transfected cells of DKO (**c**, n = 9 pairs), and DKO + T286A (**d**, n = 9 pairs). Filled circles indicate mean ± SEM (**c**, Control = 81.9 ± 9.4; DKO = 39.1 ± 6.43, p = 0.003; **d**, Control = 52.6 ± 4.9; DKO + CaMKIIα T286A = 47 ± 4.8, p = 0.31). **e** Bar graph of ratios normalized to control (%) summarizing the mean ± SEM of NMDAR EPSCs of values represented in **c** (47.8 ± 5.9, p = 0.001) and **d** (90.3 ± 7.1, p = 0.31). **f** Scatterplot shows amplitudes of AMPAR EPSCs for single pairs (open circles) of control and transfected of DKO + T286A (**d**, n = 10 pairs). Filled circle indicates mean ± SEM (**f**, Control = 153.6 ± 18.5; DKO + CaMKIIα T286A = 81.4 ± 14.8, p = 0.002). **g** Bar graph of ratios normalized to control (%) summarizing the mean ± SEM of AMPAR EPSCs of values represented in **f** (51.1 ± 4.8, p < 0.001). DKO data (red bar) from Fig. 2f are included in the graph. **h** Plots showing mean ± SEM AMPAR EPSC amplitude of control (black) and transfected (green) DKO + CaMKIIα T286A pyramidal neurons normalized to the mean AMPAR EPSC amplitude before LTP induction (arrow) (Control, n = 10; DKO + CaMKIIα T286A, n = 6, p = 0.0077 at 35 min). Sample AMPAR EPSC current traces from control (black) and electroporated neurons (green) before and after LTP are shown to the right of each graph. Gray plots represent mean ± SEM AMPAR EPSC amplitude of LTP induction in APV 50 μM. Raw amplitude data from dual cell recordings were analyzed using Wilcoxon signed rank test (p values indicated above). Normalized data were analyzed using a one-way ANOVA followed by the Mann−Whitney test (***p < 0.0001). Mann−Whitney test was also used to compare LTP at 35 min (p values indicated above). Scale bars: 50 ms, 50 pA

enhancement of AMPAR currents. Perhaps this transient phase represents a CaMKII-independent exocytosis[61] that is dependent on SNARE proteins[61,62]. This raises the possibility that some modulatory substance might be released during and immediately after induction. The present findings that LTP and sLTP are both absent from neurons lacking CaMKIIα and that exogenous constitutively active CaMKII can fully mimic LTP[15–17] confirm that CaMKII is both necessary and sufficient for NMDAR-dependent LTP. Again, there is no need to postulate the existence of CaMKII-independent processes.

It has long been known that activity translocates CaMKII from the cytosol to the PSD[40,41] and that the primary binding partner at the PSD is the GluN2B subunit of the NMDAR[38–40]. Overexpression of a GluN2B subunit in which mutations in the C-tail disrupt the binding of CaMKII to the NMDAR[42] or generation of a knockin mouse with similar mutations[43] impairs LTP. To further study the role of CaMKII binding to the NMDAR, we took advantage of two point mutations in CaMKIIα (F98K and I205K) that prevent binding to the NMDAR[40,44]. Much to our surprise these CaMKIIα mutants, which have intact kinase activity,

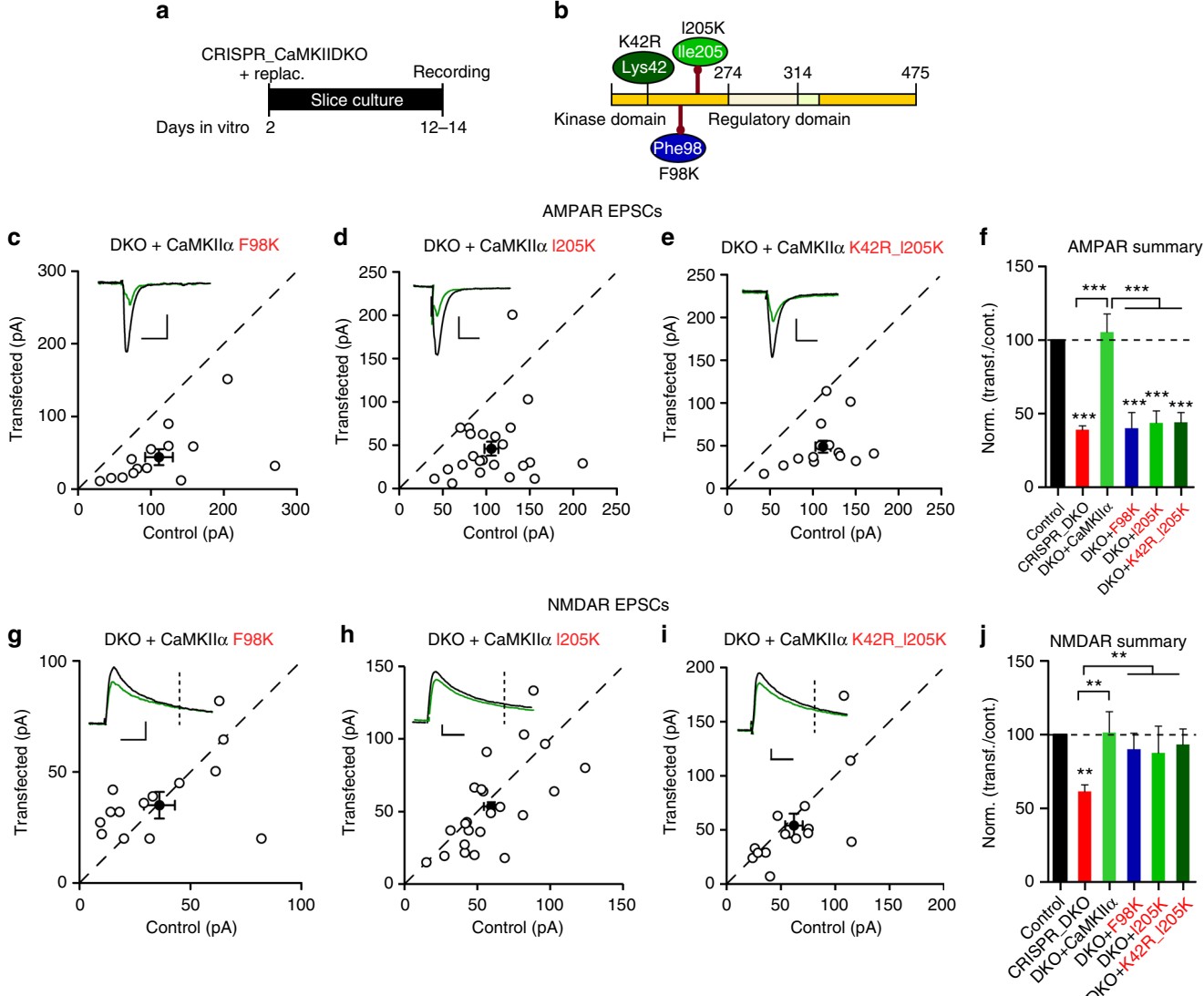

**Fig. 7** CaMKIIα binding to GluN2B is critical for basal AMPAR transmission. **a** Timeline of the experiment. **b** Scheme of the structural organization of CaMKIIα with the point mutations indicated (K42R dark green; F98K blue; I205K light green). **c**−**e** Scatterplots showing amplitudes of AMPAR EPSCs for single pairs (open circles) of control and transfected cells of DKO + CaMKIIα F98K (**c**, $n = 14$ pairs), DKO + CaMKIIα I205K (**d**, $n = 23$ pairs), and DKO + CaMKIIα K42R_I205K (**d**, $n = 14$ pairs). Filled circles indicate mean ± SEM (**c**, Control = 111 ± 19.9; DKO + CaMKIIα F98K = 44.33 ± 11.8, $p = 0.002$; **d**, Control = 106.8 ± 8.1; DKO + CaMKIIα I205K = 46.5 ± 8.6, $p < 0.0001$; **e**, Control = 112.5 ± 9; DKO + CaMKIIα I205K_K42R = 49.5 ± 7.5, $p < 0.0001$). **f** Bar graph of ratios normalized to control (%) summarizing the mean ± SEM of AMPAR EPSCs of values represented in **c** (40 ± 1.1, $p < 0.0001$), **d** (45.5 ± 7.2, $p < 0.0001$) and **e** (44.6 ± 5.6, $p < 0.0001$). DKO data (red bar) from Fig. 1l and DKO + CaMKIIα (light green) from Fig. 3e values are included in the graph. **g**−**i** Scatterplots showing amplitudes of NMDAR EPSCs for single pairs (open circles) of control and transfected cells of DKO + CaMKIIα F98K (**g**, $n = 14$ pairs), DKO + CaMKIIα I205K (**h**, $n = 23$ pairs), and DKO + K42R_I205K (**i**, $n = 14$ pairs). Filled circles indicate mean ± SEM (**g**, Control = 36.2 ± 7.7; DKO + CaMKIIα F98K = 35.3 ± 6, $p = 0.49$; **h**, Control = 59.3 ± 5.4; DKO + CaMKIIα I205K = 53.2 ± 6.4, $p = 0.20$; **i**, Control = 62.8 ± 8.5; DKO + CaMKIIα I205K_K42R = 54.9 ± 11.4, $p = 0.18$) **j** Bar graph of ratios normalized to control (%) summarizing the mean ± SEM of NMDAR EPSCs of values represented in **g** (101 ± 14, $p = 0.62$), **h** (91.1 ± 7.3, $p = 0.74$) and **i** (88.2 ± 10.2, $p = 0.66$). DKO data (red bar) from Fig. 1p and DKO + CaMKIIα (light green bar) from Fig. 3h values are included in the graph. Raw amplitude data from dual cell recordings were analyzed using Wilcoxon signed rank test ($p$ values indicated above). Normalized data were analyzed using a one-way ANOVA followed by the Mann−Whitney test (***$p < 0.0001$; **$p < 0.001$). Scale bars: 50 ms, 50 pA

behaved identically to CaMKIIα T286A and K42R: they failed to rescue AMPAR EPSCs in the DKO background, but rescued the defect in NMDAR transmission. Furthermore, LTP was absent in the CaMKIIα I205K mutant (Fig. 8). Perhaps even more surprising is the finding that overexpression of the constitutively active kinase (CaMKIIα T286D-T305A/T306A), which enhances AMPAR responses three-fold, no longer had any effect on synaptic responses when it harbored the I205K mutation. This finding indicates that for CaMKII to function at the synapse it

must be targeted to a precise nanodomain by the NMDAR, which thus presumably restricts its spatial influence.

Why is the disruption of CaMKIIα binding to the NMDAR in our experiments using the CaMKIIα F98K and I205K mutations more dramatic compared to disrupting binding with mutations in the GluN2B C-terminal domain[42,43]? In addition to the difference in experimental conditions, it is possible that the CaMKIIα F98K and I205K mutations might also disrupt CaMKII binding to the GluN1 subunit[47] and other additional synaptic proteins[45]. To

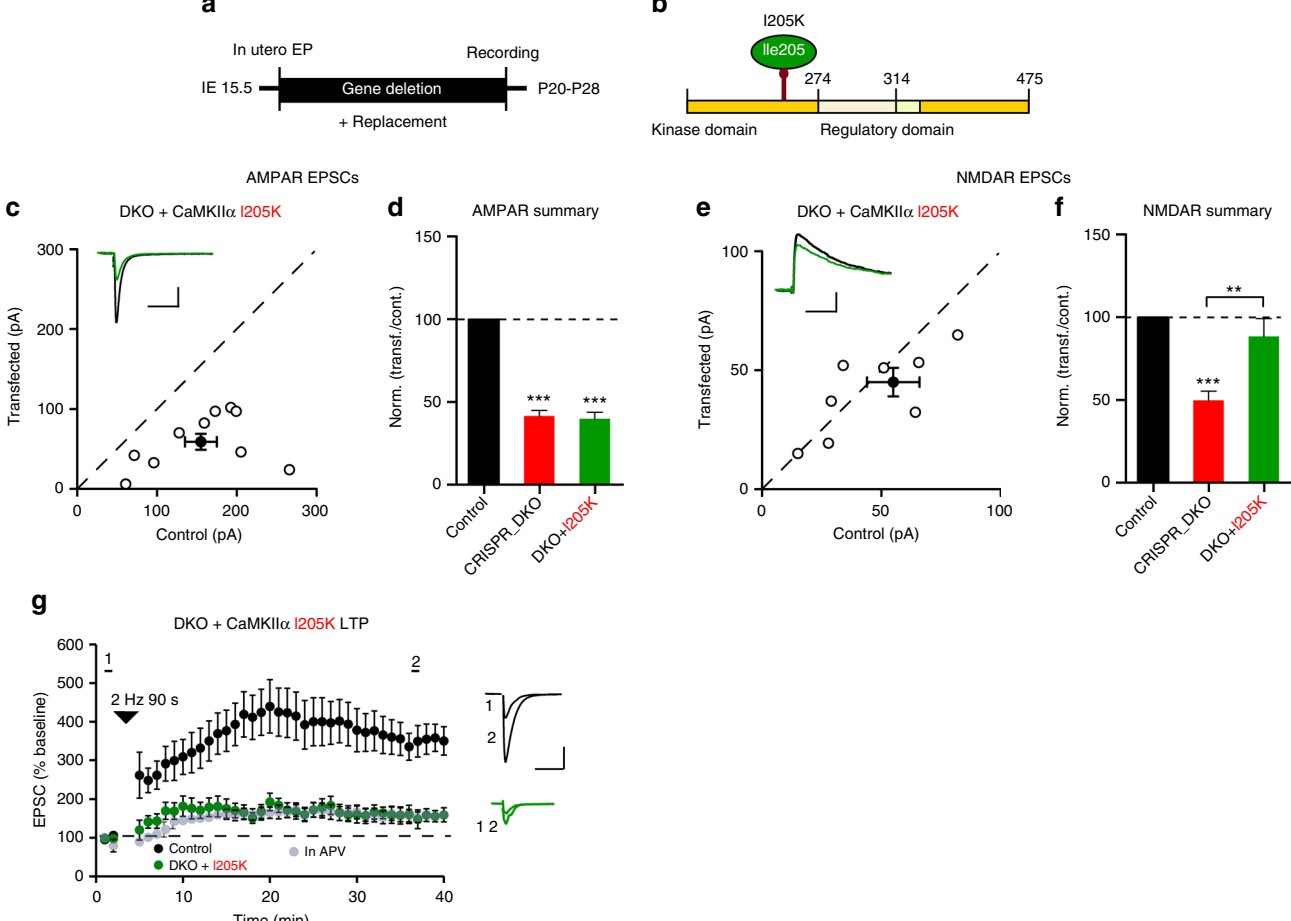

**Fig. 8** CaMKIIα binding to GluN2B is critical for LTP. **a** Timeline of the experiment. **b** Scheme of the structural organization of CaMKIIα showing the point mutation (I205K light green). **c** Scatterplot showing amplitudes of AMPAR EPSCs for single pairs (open circles) of control and transfected cells of DKO + CaMKIIα I205K (**c**, n = 10 pairs). Filled circle indicates mean ± SEM (**c**, Control = 155.2 ± 20.7; DKO + CaMKIIα I205K = 59.8 ± 10.8, p < 0.0001). **d** Bar graph of ratios normalized to control (%) summarizing the mean ± SEM of AMPAR EPSCs of values represented in **c** (39.9 ± 6.1, p = 0.0007). DKO data (red bar) from Fig. 2f are included in the graph. **e** Scatterplot showing amplitudes of NMDAR EPSCs for single pairs (open circles) of control and transfected cells of DKO + CaMKIIα I205K (**d**, n = 9 pairs). Filled circle indicates mean ± SEM (**e**, Control = 55.3 ± 11.7; DKO + CaMKIIα I205K = 43.4 ± 6.2, p = 0.17). **f** Bar graph of ratios normalized to control (%) summarizing the mean ± SEM of NMDAR EPSCs of values represented in **e** (90.3 ± 11.2, p = 0.38). DKO data (red bar) from Fig. 6e are included in the graph. **g** Plots showing mean ± SEM AMPAR EPSC amplitude of control (black) and transfected (green) CRISPR_DKO + CaMKIIα I205K pyramidal neurons normalized to the mean AMPAR EPSC amplitude before LTP induction (arrow) (Control, n = 12; DKO + CaMKIIα I205K, n = 9, p = 0.0049 at 35 min). Sample AMPAR EPSC current traces from control (black) and electroporated neurons (green) before and after LTP are shown to the right of each graph. Gray plots represent mean ± SEM AMPAR EPSC amplitude of LTP induction in APV 50 μM. Raw amplitude data from dual cell recordings were analyzed using Wilcoxon signed rank test (p values indicated above). Normalized data were analyzed using a one-way ANOVA followed by the Mann−Whitney test (***p < 0.0001; **p < 0.001). Mann−Whitney test was used to compare LTP at 35 min (p values indicated above). Scale bars: 50 ms, 50 pA

address this possibility we replaced GluN2B with a mutant that is unable to bind to CaMKII. This mutant mimicked the effects of deleting CaMKIIα, confirming that binding of CaMKIIα to the NMDAR is, indeed, crucial for its synaptic actions. The NMDARs tend to be clustered in the center of the synapse, whereas the AMPARs are more concentrated around the periphery[63,64]. The apparent requirement for CaMKII to be bound to the NMDAR to recruit AMPARs to the synapse suggests that CaMKII exerts its effects well beyond the center of the synapse. This might be explained by the dynamic nature of synaptic proteins, such as AMPARs and scaffolding proteins. In addition, CaMKII binding to the NMDAR initiates a widespread remodeling of the PSD, most likely involving the actin cytoskeleton[58,65]. Another issue concerns stoichiometry. There are, on average, about 20 NMDARs per synapse compared to about 80−170 CaMKII molecules per synapse[66,67]. Hence, it seems likely that CaMKII

interacts with additional PSD proteins, although the physiological consequences of these interactions are unclear.

One of the more controversial topics in the LTP field concerns the mechanism underlying its persistence. It has long been postulated that the "memory" is due to the unique biochemical properties of the CaMKII protein, in which the transient $Ca^{2+}$/CaM activation of CaMKII initiates its autophosphorylation converting the kinase into a $Ca^{2+}$-independent constitutively active form[1,6,8]. The fact that we observe a large decrease in baseline AMPAR transmission following the deletion of CaMKII would be consistent with this model. However, other experiments have raised serious concerns. If constitutive CaMKII activity is responsible for LTP maintenance one would expect that blockade of CaMKII activity after the induction of LTP should reverse the potentiation. However, this is generally not the case[9,11,57,68], but see ref. [24]. To examine this issue we took advantage of CaMKIIα

T305A/T306A mutations, which makes CaMKII highly sensitive to $Ca^{2+}$/CaM. Expression of this construct caused a three-fold enhancement in AMPAR currents. One might expect this enhancement to be initially driven by $Ca^{2+}$/CaM, but, via phosphorylation of T286, CaMKII should become constitutively active. We found that application of APV reversed the enhancement indicating that spontaneous NMDAR activation drives the enhancement. Importantly, application of APV for as short a time as an hour entirely reversed the enhancement. Although we could find no evidence for a $Ca^{2+}$/CaM-independent constitutive component to the CaMKII enhancement, these experiments do not exclude such a mechanism. As proposed by Lisman et al.[69], CaMKII is postulated to assume three distinct activation states that depend on the magnitude, and possibly the duration, of the $Ca^{2+}$ signal. The activation of CaMKII with weak

signals fails to initiate autophosphorylation and the kinase inactivates rapidly (0.1–0.2 s) as calmodulin dissociates from the kinase. This would explain the moment-to-moment signaling of NMDARs in the absence of any lasting changes[70]. With modest $Ca^{2+}$ elevation autophosphorylation will occur. The duration of the signal is proposed to depend on the phosphatase activity, which can dephosphorylate CaMKII over a time scale of minutes[71]. Finally, it is proposed that with robust elevation in $Ca^{2+}$, as occurs with LTP, the rate of autophosphorylation is greater than the rate of dephosphorylation. A conservative interpretation of our findings with the T305A/T306A mutation is that the level of CaMKII activation fails to reach threshold for favoring autophosphorylation over dephosphorylation.

Our work emphasizes the central role that CaMKIIα plays in excitatory synaptic transmission. It is required both in early

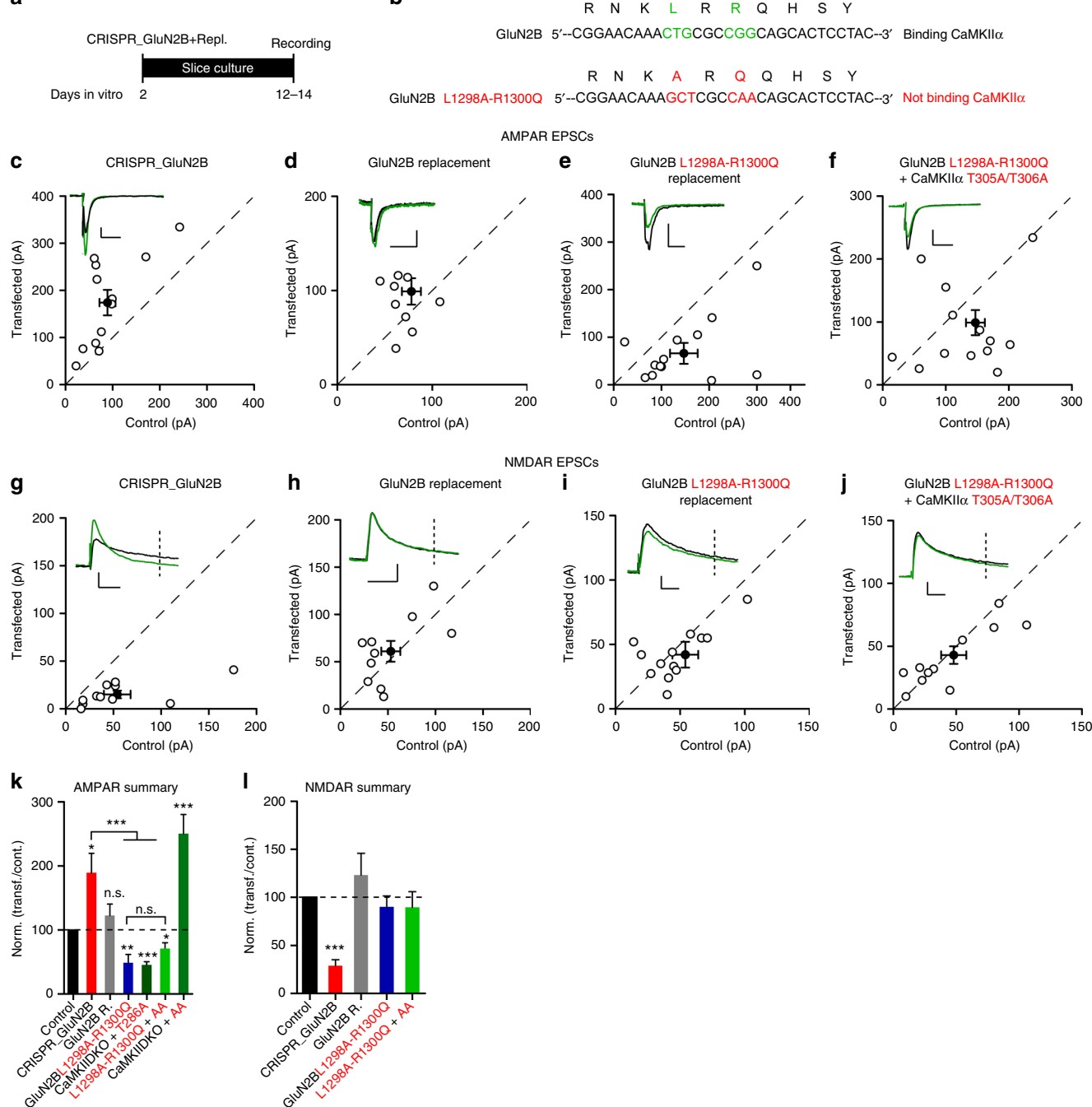

**Fig. 9** Replacement of GluN2B that does not bind CaMKIIα confirms the role of this interaction in basal transmission. **a.** Timeline of experiment. **b** Scheme of GluN2B region critical for CaMKII binding (green wild type; red mutated). **c–f** Scatterplots showing amplitudes of AMPAR EPSCs for single pairs (open circles) of control and transfected cells of CRISPR_GluN2B (**c**, $n = 12$ pairs), CRISPR_GluN2B + GluN2B (**d**, $n = 10$), CRISPR_GluN2B + GluN2Bmut (**e**, $n = 14$ pairs), and CRISPR_GluN2B + GluN2Bmut + CaMKIIα T305A/T306A (**f**, $n = 13$ pairs). Filled circles indicate mean ± SEM (**c**, Control = 89.38 ± 17.50; CRISPR_GluN2B = 174 ± 27.83, $p = 0.01$; **d**, Control = 78.9 ± 10.8; CRISPR_GluN2B + GluN2B = 99.3 ± 14.8, $p = 0.28$; **e**, Control = 148.4 ± 22.7; CRISPR_GluN2B + GluN2Bmut = 75.8 ± 18.6, $p = 0.01$; **f**, Control = 139.0 ± 16.6; CRISPR_GluN2B + GluN2Bmut + CaMKIIα T305A/T306A = 92 ± 20.2, $p = 0.07$). **g–j** Scatterplots showing amplitudes of NMDAR EPSCs for single pairs (open circles) of control and transfected cells of CRISPR_GluN2B (**g**, $n = 12$ pairs), CRISPR_GluN2B + GluN2B (**h**, $n = 10$ pairs), CRISPR_GluN2B + GluN2Bmut (**i**, $n = 14$ pairs), and CRISPR_GluN2B + GluN2Bmut + CaMKIIα T305A/T306A (**j**, $n = 13$ pairs). Filled circles indicate mean ± SEM (**g**, Control = 54.7 ± 14.4; CRISPR_GluN2B = 15.5 ± 4, $p = 0.016$; **h**, Control = 53.2 ± 10.2; CRISPR_GluN2B + GluN2B = 61.8, $p = 0.58$; **i**, Control = 54.15 ± 10.87; CRISPR_GluN2B + GluN2Bmut = 42.5 ± 10.4, $p = 0.45$; **j**, Control = 49.5 ± 9.3; CRISPR_GluN2B + GluN2Bmut + CaMKIIα T305A/T306A = 41.4 ± 6.7, $p = 0.48$). **k** Bar graph of ratios normalized to control (%) summarizing the mean ± SEM of AMPAR EPSCs of values represented in **c** (180 ± 31.1, $p = 0.02$), **d** (129 ± 20.4, $p = 0.22$), **e** (48.5 ± 13.3, $p = 0.005$) and **f** (68.3 ± 14.6, $p = 0.04$). Values for CaMKIIα T286A (Fig. 5e) and CaMKIIα T305A/T306A (Suppl. Figure 10) replacements are included (% of control: 45.4 ± 5 and 250 ± 30). **l** Bar graph of ratios normalized to control (%) summarizing mean ± SEM of NMDAR EPSCs of values represented in **g** (28.8 ± 14.8, $p = 0.0002$), **h** (123 ± 22.6, $p = 0.60$), **i** (90.3 ± 11.1, $p = 0.98$) and **j** (89.9 ± 16, $p = 0.74$). Raw amplitude data from dual cell recordings were analyzed using Wilcoxon signed rank test ($p$ values indicated above). Normalized data were analyzed using a one-way ANOVA followed by the Mann−Whitney test (\*\*\*$p < 0.0001$; \*\*$p < 0.001$; \*$p < 0.01$). Scale bars: 50 ms, 50 pA

development and in the adult to maintain basal synaptic AMPARs via its kinase activity and NMDARs via a kinase-independent mechanism. CaMKIIα can fully account for NMDAR-dependent LTP. CaMKIIβ plays no role in maintaining basal synaptic transmission, but can modulate CaMKIIα-dependent LTP. sLTP involves a fast-transient spine enlargement which is independent of CaMKII, but dependent on $Ca^{2+}$, followed by a persistent CaMKII-dependent enlargement. We were unable to identify a $Ca^{2+}$/CaM-independent constitutive component to the CaMKII enhancement. Finally all the synaptic effects of CaMKIIα that we examined, except the maintenance of baseline NMDAR currents, required its binding to NMDARs, establishing the CaMKII/NMDAR protein complex as an essential synaptic signaling hub, controlling numerous fundamental aspects of excitatory synaptic transmission.

## Methods

**DNA constructs**. Design and screening of gRNAs for the CRISPR constructs were performed as previously described[53]. Briefly for the screening of CaMKIIα and β gRNA sequences for potential off-target effects, we used the Cas9 design target tool (http://crispr.mit.edu). The human codon-optimized Cas9 and chimeric gRNA expression plasmid (px458) as well as the lentiviral backbone plasmid (lentiCRISPR v.2) both developed by the Zhang lab[72–74] were obtained from Addgene. Lenti-CRISPR was modified inserting an EGFP sequence after a p2a promoter to detect the expression level after infection. After screening several gRNAs the primers used to design the specific gRNA targets were: CaMK2a#1 forward (5′ to 3′) CACC G ctccaggggagccttctccg; CaMK2a#1 reverse (3′ to 5′) AAAC cggagaaggctcccctggag C; CaMK2a#2 forward (5′ to 3′) CACC Gcaggtgatggtagccatcc; CaMK2a#1 reverse (3′ to 5′) AAAC ggatggctaccatcacctgC; CaMK2b#1 forward (5′ to 3′) CACC Gtgtcatggaggcgtactgcg; CaMK2b#1 reverse (3′ to 5′) AAAC cgcagtacgcctccatgacaC. All gRNAs were accurately chosen to target both mouse and rat genes. In particular CaMK2a#1 targets a region in which the PAM sequence is in an intron of the catalytic region. CaMK2a#2 targets part of the UTR region of exon 1. CaMK2b#1 gRNA targets a region between intron 1 and exon 1. Thus all gRNAs are not targeting the coding region of the CaMKII plasmid, allowing replacements with wt constructs. pRSV-CaMKIIα was a gift from Richard Maurer (Addgene plasmid # 45064) and GFP-C1-CaMKIIβ was a gift from Tobias Meyer (Addgene plasmid # 21227). Both plasmids were sub-cloned into a pCAGGS vector followed by an IRES-GFP or mCherry sequence for biolistic transfection. All CaMKIIα mutations were done by overlapping PCR from the pCAGGS-CaMKIIα−IRES-GFP or mCherry. For all replacement experiments to target both CaMKII isoforms we sub-cloned the two gRNAs including the following chimeric RNA into a pFUGW mCherry or GFP plasmid, through PCR amplification and insertion into the FUGW plasmid using *Bst*bI and *Eco*RI restriction sites. Primers for CaMK2a gRNA were: forward TTAATCGTACGAATTCgagggcctatttccc; reverse GGGTTAATTAATTCGAATGG CGTTCTATTGA. Primers for CaMK2b gRNA: forward TAGTAACGCCATTGC AAgagggcctatttccc; reverse GGGTTAATTAATTCGAATGGCGTTACTATTGA. All cloning was achieved with In-Fusion HD Cloning System (Clontech). For over-expression experiments a pFUGW vector expressing only GFP was coexpressed with pCAGG-IRES-mCherry constructs to enhance the identification of transfected neurons, and served as a control vector for spine imaging.

**Sequence of CaMKIIα, β and GluN2B used in all replacements and primers to make the mutations**. Amino acidic sequence of the CaMKIIα (*Rattus Norvegicus*) used in all experiments:

MATITCTRFTEEYQLFEELGKGAFSVVRRCVKVLAGQEYAAKIINTKKLSA RDHQKLEREARICRLLKHPNIVRLHDSISEEGHHYLIFDLVTGGELFEDIVARE YYSEADASHCIQQILEAVLHCHQMGVVHRDLKPENLLLASKLKGAAVKLAD FGLAIEVEGEQQAWFGFAGTPGYLSPEVLRKDPYGKPVDLWACGVILYILLVG YPPFWDEDQHRLYQQIKAGAYDFPSPEWDTVTPEAKDLINKMLTINPSKRIT AAEALKHPWISHRSTVASCMHRQETVDCLKKFNARRKLKGAILTTMLATRN FSGGKSGGNKKNDGVKESSESTNTTIEDEDTKVRKQEIIKVTEQLIEAISNGDF ESYTKMCDPGMTAFEPEALGNLVEGLDFHRFYFENLWSRNSKPVHTTILNPH IHLMGDESACIAYIRITQYLDAGGIPRTAQSEETRVWHRRDGKWQIVHFHRS GAPSVLPH.

All mutations were done from this sequence using overlapping PCR of a pCAGGS-CaMKIIα in a pCAGGS-IRES-GFP or mCherry plasmid digested with *Nhe*I and *Xho*I. Common outside primers were: forward (5′ to 3′) 5′-GGACTCA GATCTCGAGATGGCTACCATCACC-3′; Reverse (3′ to 5′) GAAGCTTGAGCT CGAGTCAATGGGGCAGGAC. The following primers were used for the K42R mutation: inside forward GAGTATGCTGCCAGGATTATCAACAC; inside reverse GAGTATGCTGCCAGGATTATCAACAC. For T286A mutation: inside forward GACAGGAGGCAGTGGACTGCCTGAAG; inside reverse CTTCAGGCAGTCCACTGCCTCCTGTC. For T286D: inside forward GACAGGAGGACGTGGACTGCCTGAAG; inside reverse CAGTCCACGTCCTCCTGTCTGTGCAT. For I205K: inside forward GGCGTCATCCTGTAAATCTTGCTGGTT; inside reverse: AACCAGCAAGATTTACAGGATGACGCC. For F98K: inside forward: TGGTGGGGAGCTGAAAGAAGACATTGT; inside reverse: AACCAGCAAGATTTACAGGATGACGCC. For T305A/T306A: inside forward GCCATCCTCGCCGCTATGCTGGCCACC; inside reverse ACAATGTCTTCTTTCAGCTCCCCACCA. For T305D/T306D; inside forward GCCATCCTCGATGACATGCTGGCCACC; inside reverse GGTGGCCAGCATGTCATCGAGGATGGC. For T305A/T306D: inside forward GCCATCCTCGATGACATGCTGGCCACC; inside reverse GGTGGCCAGCATGTCATCGAGGATGGC. For T305D/T06A: inside forward GCCATCCTCGACGCCATGCTGGCCACC; inside reverse GGTGGCCAGCATGGCGTCGAGGATGGC. Some of the mutations were combined (e.g. T286D-T305A/T306A) using the first mutated form as backbone.

Amino acidic sequence of the CaMKIIβ (isoform 2 *Rattus Norvegicus*) used in the experiments: MATTVTCTRFTDEYQLYEDIGKGAFSVVRRCVKLCTGHEYA AKIINTKKLSARDHQKLEREARICRLLKHSNIVRLHDSISEEGFHYLVFDLVTG GELFEDIVAREYYSEADASHCIQQILEAVLHCHQMGVVHRDLKPENLLLASK CKGAAVKLADFGLAIEVQGDQQAWFGFAGTPGYLSPEVLRKEAYGKPVDIW ACGVILYILLVGYPPFWDEDQHKLYQQIKAGAYDFPSPEWDTVTPEAKNLIN QMLTINPAKRITAHEALKHPWVCQRSTVASMMHRQETVECLKKFNARRKLK GAILTTMLATRNFSVGRQTTAPATMSTAASGTTMGLVEQAKSLLNKKADG VKPQTNSTKNSSAITSPKGSLPPAALEPQTTVIHNPVDGIKESSDSTNTTIEDED AKARKQEIIKTTEQLIEAVNNGDFEAYAKICDPGLTSFEPEALGNLVEGMDFH RFYFENLLAKNSKPIHTTILNPHVHVIGEDAACIAYIRLTQYIDGQGRPRTSQS EETRVWHRRDGKWQNVHFHCSGAPVAPLQ,

For GluN2B replacement experiments overlapping PCR of a p-CAGGS-GluN2B-GFP plasmid was used. In particular, mutated GluN2B was inserted in a p-CAGGS-IRES-mCherry plasmid digested with *Nhe*I and *Xho*I. Outside primers, forward: ATTCGCGGCCGCTAGCATGAAGCCCCAGCGC; reverse: GAAGCTT GAGCTCGAGTCAGACATCAGACTC. Inside primers, forward: GAAGAATCG GAACAAAGCTCGCCAACAGCACTCCTACGACA; reverse: TGTCGTAGGAG TGCTGTTGGCGAGCTTTGTTCCGATTCTT.

**Slice culture, transfection, and CaMKII molecular replacement**. Organotypic hippocampal slice cultures were prepared from P6-8 rat brains and transfected with sparse biolistic transfections as previously described[75,76]. Briefly, 50 µg of each plasmid DNA was coated on 1 µm diameter gold particles in 0.5 mM spermidine, precipitated with 0.1 mM CaCl$_2$, and washed four times in pure ethanol. The gold particles were coated onto PVC tubing, dried using ultra-pure N2 gas, and stored at 4 °C in desiccant. DNA-coated gold particles were delivered with a Helios Gene Gun (BioRad). Constructs were transfected at DIV2. After 10–12 days of transfection, neurons transfected with CRISPR constructs could be identified by GFP or mCherry epifluorescence.

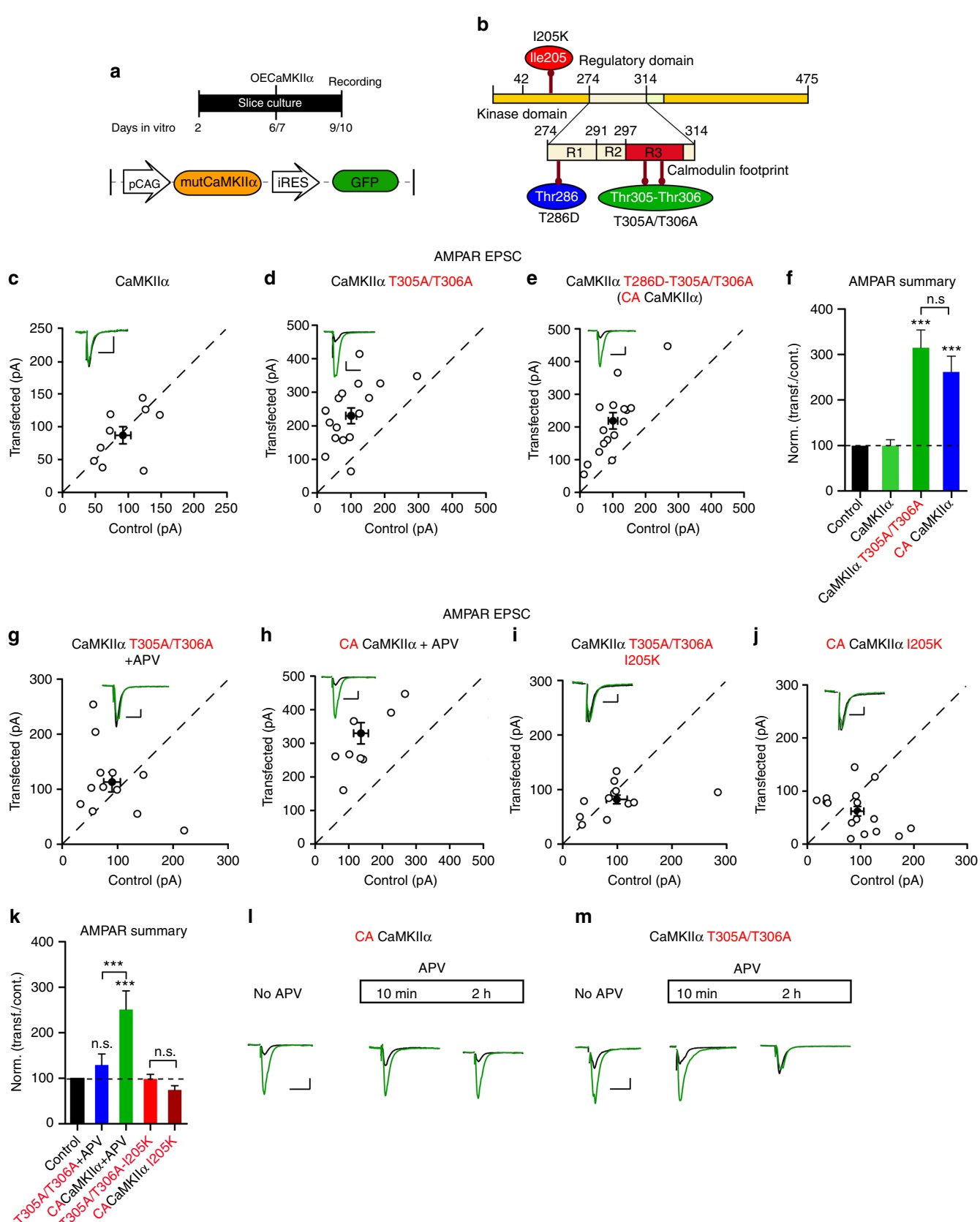

**Fig. 10** NMDAR-dependent and independent constitutive actions of CaMKIIα. **a** Timeline of the experiment. **b** Scheme of the structural organization of CaMKIIα showing the point mutations (I205K red; T286D blue; T305A/T306A green). **c**−**e** Scatterplots showing amplitudes of AMPAR EPSCs for single pairs (open circles) of control and transfected cells of overexpressed WT CaMKIIα (**c**, $n = 9$ pairs), sensitive CaMKIIα T305A/T306A (**d**, $n = 17$ pairs) and constitutively active CaMKIIα T286D-T305A/T306A (**e**, $n = 17$ pairs). Filled circles indicate mean ± SEM (**c**, Control = 92.4 ± 12.3; CaMKIIα = 87.5 ± 13.9, $p = 0.83$; **d**, Control = 101.2 ± 16.3; CaMKIIα T305A/T306A = 230.7 ± 23.4, $p = 0.0005$; **e**, Control = 101.5 ± 14.1; CaMKIIα T286D-T305A/T306A = 219.1 ± 26, $p = 0.0005$). **f** Bar graph of ratios normalized to control (%) summarizing the mean ± SEM of AMPAR EPSCs of values represented in **c** (99 ± 13 $p = 0.59$), **d** (309.8 ± 5.3, $p < 0.0001$) and **e** (249.8 ± 26.3, $p = 0.001$). **g**−**j** Scatterplots showing amplitudes of AMPAR EPSCs for single pairs (open circles) of control and transfected cells of overexpressed sensitive CaMKIIα T305A/T306A (**g**, $n = 12$ pairs) and constitutive active CaMKIIα T286D-T305A/T306A (**h**, $n = 8$ pairs), after 50 µM APV treatment; CaMKIIα T305A/T306A_I205K (**i**, $n = 11$ pairs) and CaMKIIα T286D-T305A/T306A_I205K (**j**, $n = 15$ pairs). Filled circles indicate mean ± SEM (**g**, Control = 100 ± 15.3; CaMKIIα T305A/T306A = 113.6 ± 18.4, $p = 0.27$; **h**, Control = 124.2 ± 22.6; CaMKIIα T286D -T305A/T306A = 296.9 ± 32, $p = 0.007$; **i**, Control = 99.3 ± 19.8; CaMKIIα T305A/T306A_I205K = 82.1 ± 8.3, $p = 0.38$; **j**, Control = 94.7 ± 12.2; CaMKIIα T286D-T305A/T306A_I205K = 62.9 ± 10, $p = 0.10$). **k** Bar graph of ratios normalized to control (%) summarizing the mean ± SEM of AMPAR EPSCs of values represented in **g** (150.7 ± 30, $p = 0.18$), **h** (270.4 ± 39, $p < 0.0001$), (**i**) (100.2 ± 13.8, $p = 0.35$) and **j** (93.3 ± 20.3, $p = 0.67$ Mann−Whitney test). **l** Sample traces of AMPAR EPSCs of control (black) and CA CaMKIIα transfected (green) neurons before and after APV treatment. **m** Sample traces of AMPAR EPSCs of control (black) and CaMKIIα T305A/T306A transfected (green) neurons before and after APV treatment. Raw amplitude data from dual cell recordings were analyzed using Wilcoxon signed rank test ($p$ values indicated above). Normalized data were analyzed using a one-way ANOVA followed by the Mann−Whitney test (\*\*\*$p < 0.0001$; \*\*$p < 0.001$; \*$p < 0.01$). Scale bars: 50 ms, 50 pA. See also Supplementary Figs. 9 and 10

**In utero electroporation**. In utero electroporation was performed as previously described[77]. E15.5 pregnant WT (CD-1) mice were anesthetized with 2% isoflurane in $O_2$ and buprenorphine (Reckitt Benckiser Healthcare) and meloxicam (Boehringer Ingelheim) were used for analgesia. Embryos were exposed and injected with ~1.5 µl of mixed plasmid DNA with Fast Green (Sigma Aldrich) into the lateral ventricle via a beveled micropipette. Embryos were exposed and injected with ~2 µg/µl of the pFUGW-CaMKIIDKO-mCherry, px458 and pCAGGS- plasmids. Each embryo was electroporated with five 40 V pulses of 50 ms, delivered at 1 Hz, using platinum tweezertrodes in a square-wave pulse generator (BTX Harvard Apparatus). The positive electrode was placed in the lower right hemisphere and the negative electrode placed in the upper left hemisphere. Following electroporation, the embryos were placed back into the abdominal cavity and abdominal muscle and skin were sutured. All experiments were performed in accordance with established protocols approved by the University of California, San Francisco's Institutional Animal Care and Use Committee.

**Lentivirus production**. Three T-75 flasks of rapidly dividing HEK293T cells (ATCC) were transfected with 27 µg lentiCRISPR v.2-GFP for single CaMKII isoforms gRNAs or pFUGW-CaMKIIDKO-mCherry, plus helper plasmids pVSV-G (18 µg) and psPAX2 (27 µg) using FuGENE HD (Promega). DNA was incubated with 210 µL FuGENE HD in 4.5 ml Opti-MEM (Life Technologies) before transfection, according to the manufacturer's directions. Forty hours later, supernatant was collected, filtered, and concentrated using the PEG-it Virus Precipitation Solution (System Biosciences) according to the manufacturer's directions. The resulting pellet was resuspended in 400 µL Opti-MEM or PBS, flash-frozen, and stored at −80 °C.

**Neuronal culture transduction and western blot**. Dissociated cultures of postnatal rat hippocampal neurons were transduced with freshly made viral supernatant at DIV4. For western blot analysis, neurons (DIV18) were lysed in 150 mM NaCl, 50 mM Tris-HCl, pH 8.0, 1% TX-100, and protease inhibitors (Roche) including 1 mM EGTA and 1 mM PMSF. After sedimentation at 14,000 g to remove nuclei and cell debris, 5 µg protein was separated by electrophoresis through polyacrylamide, transferred to nitrocellulose, and the membranes immunoblotted for CaMKIIα (1 µg/mL, Enzo, monoclonal 6G9), CaMKIIβ (1 µg/mL, Abcam ab34703), β-actin (1 µg/mL, Millipore 4C2), β-tubulin (1 µg/mL, Millipore AA2) and the appropriate secondary antibodies conjugated to IRDye800 (Rockland). The membrane was imaged with a LICOR system (Odyssey).

**Adult mice viral injections**. For adult CaMKII DKO experiments Rosa26-Cas9 knock-in mice were used. Mice of both sexes with FVB/NJ congenic background were purchased from Jackson labs (Stock No. 026558). These mice constitutively express Cas9 and GFP. Viral injections of the pFUGW-CaMKIIDKO-mCherry lentivirus were performed at P48 to P56. Mice were anesthetized with isoflurane and immobilized on a stereotaxic frame. Bilateral injections of lentiviral vectors into the CA1 Stratum Radiatum (AP: −2.00 mm, ML: +/−1.50 mm, DV: −1.50 mm, relative to Bregma), were conducted delivering 250 nL per hemisphere at a rate of 500 nL/min via a Hamilton 88011 syringe driven by a Micro4 microsyringe pump (World Precision Instruments). All experiments were performed in accordance with established protocols approved by the University of California, San Francisco's Institutional Animal Care and Use Committee. Acute hippocampal slices were prepared for recording 21 days later. Transfected pyramidal neurons in area CA1 were identified by morphology and location. Some slices presented transfection in the CA3, which we used as a proof of concept to study the importance of CaMKII in a different neuronal type.

**Electrophysiology**. Whole-cell recordings in area CA1 were done by simultaneously recording responses from a fluorescent transfected neuron and a neighboring untransfected control neuron. Pyramidal neurons were identified by morphology and location. Series resistance was monitored online, and recordings in which series increased to >30 MΩ or varied by >50% between neurons were discarded. Dual whole-cell recordings measuring evoked EPSCs used an extracellular solution bubbled with 95% $O_2$/ 5% $CO_2$ consisting of (in mM) 119 NaCl, 2.5 KCl, 4 $CaCl_2$, 4 $MgSO_4$, 1 $NaH_2PO_4$, 26.2 $NaHCO_3$, 11 Glucose. 100 µM picrotoxin was added to block inhibitory currents and 0–2 µM 2-Chloroadenosine was used to control epileptiform activity in organotypic slices. Intracellular solution contained (in mM) 135 CsMeSO$_4$, 8 NaCl, 10 HEPES, 0.3 EGTA, 5 QX314-Cl, 4 MgATP, 0.3 $Na_3GTP$, 0.1 spermine. A bipolar stimulation electrode (FHC) was placed in stratum radiatum, and responses were evoked at 0.2 Hz. Peak AMPAR currents were recorded at −70 mV. NMDAR currents measured at +40 mV and were temporally isolated by measuring amplitudes 100 ms following the stimulus. Paired-pulse ratio was determined by delivering two stimuli 40 ms apart and dividing the peak response to stimulus 2 by the peak response to stimulus 1. mEPSCs were isolated by adding 0.5–1 µM TTX to the recording solution to block evoked potentials and were analyzed off-line with custom software (IGOR Pro). To minimize runup of baseline responses during LTP, cells were held cell-attached for 1–2 min before breaking into the cell. LTP was induced by holding neurons at 0 mV during a 2-Hz stimulation of Schaffer collaterals for 90 s. Dual whole-cell recordings in Dentate Gyrus were done by simultaneously recording responses from a fluorescent transfected neuron and neighboring untransfected control neuron.

**2P microscopy imaging and measurements of sLTP in single spines**. Hippocampal slices were biolistically transfected with pFUGW-GFP for control, pFUGW-CaMKIIDKO-GFP, Cas9 (px 330) for DKO and pFUGW-CaMKIIDKO-GFP, Cas9 (px 330) and pCAGSS-CaMKIIαT286A-mCherry for replacement. Slices were maintained at room temperature (r.t. 25–27 °C) in a continuous perfusion of Mg-free artificial cerebrospinal fluid (ACSF) containing (in mM): 119 NaCl, 2.5 KCl, 4 $CaCl_2$, 26.2 $NaHCO_3$, 1 $NaH_2PO_4$ and 11 glucose, 1 µM tetrodotoxin, 50 µM picrotoxin and 4-methoxy-7-nitroindolinyl-caged L-glutamate (MNI-glutamate, 2.5 mM, Tocris), equilibrated with 5% $CO_2$/95% $O_2$. Imaging was performed at 10−14 DIV in primary or secondary dendrites from the distal part of the main apical dendrite of CA1 pyramidal neurons. Neurons were visualized using a two-photon imaging system (Bruker, Ultima II) powered by two Ultra II femtosecond lasers (Coherent). Neurons were imaged with a 910 nm excitation source, and GFP fluorescence was collected on R9110 photomultiplier tubes in epi- and transfluorescence configurations, downstream of 550/100 bandpass filters. Scanning interference contrast images of slice morphology was acquired with a photomultiplier tube downstream of a 770 nm longpass filter to ensure that all uncaging sites were within 15 µm of the slice surface.

To induce sLTP, MNI glutamate was photolyzed using a 720 nm excitation source (7.5 mW at the focal point). sLTP was induced by 0.5 ms pulses at 0.5 Hz, with uncaging pulses positioned close to the tip of the spine. Images were obtained at the Nyquist resolution limit, with four images averaged every second. For the long-term sLTP, a z-stack image every 30 s was taken in order to have a precise reconstruction of the spine changes. All analysis was done using Fiji (ImageJ) software. For analysis two different methods were used: (i) for short-term analysis the spine of interest was selected by a circle and mean fluorescence was measured using ROI manager multi measure. The values were normalized to the first 15 frames of baseline ($F_0$) before the uncaging. (ii) For long-term analysis, head diameter was obtained using full width tenth-maximum (FWTM) measurements based on Gaussian fits to approximate manual head measurement, using the curve fitter plug-in. The spine head diameter was assessed every 30s from a single optical

section within the z-stack that best transected the head. Data were then processed with Excel software and shown as graphs with GraphPad.

**Spine morphology analysis**. For spine morphology measurements, images were acquired at DIV 7-9 using super-resolution microscopy (N-SIM Microscope System, Nikon). For use with the available inverted microscope and oil-immersion objective lens, slices were fixed in 4% PFA/4% sucrose in PBS and washed 3× with PBS. To amplify the GFP signal, slices were then blocked and permeablized in 3% BSA in PBS containing 0.1% Triton-X and stained with primary antibody against GFP (2 μg/mL, Life Technologies A-11122) followed by washes in PBSTx and staining with Alexa 488-conjugated secondary (4 μg/mL, Life Technologies A-11034). Slices were mounted in SlowFade Gold (Life Technologies) for imaging. Only dendrites in the top 20 μm of the slice were imaged. Some slices were further processed with an abbreviated SeeDB-based protocol[78] in an attempt to reduce spherical aberration, but no substantial improvement was seen. Images were acquired with a ×100 oil objective in 3D-SIM mode using supplied SIM grating (3D EX V-R ×100/1.49) and processed and reconstructed using supplied software (NIS-Elements, Nikon). Morphological analysis was done on individual sections using ImageJ to perform geometric measurements on spines extending laterally from the dendrite. Spine neck widths were obtained from full width half-maximum measurements based on Gaussian fits of line profile plots[79]. Neck length was measured from the base of the spine to the base of the head. Head diameter was measured perpendicular to the spine neck axis through the thickest part of the spine head. Head diameter was obtained using full width tenthmaximum (FWTM) measurements based on Gaussian fits to approximate manual head measurement. We note that others[79] have recently performed more precise morphological measurements; we are constrained by available tools and tissue preparation.

**Live confocal imaging**. CA1 pyramidal neurons in organotypic hippocampal slice cultures made from P6 rat pups were biolistically transfected with GFP or mCherry tagged constructs ~18–20 h after plating. Confocal imaging was performed on live tissue in HEPES-buffered ACSF (125 mM NaCl; 5 mM KCl; 10 mM D-Glucose; 10 mM HEPES; 2 mM MgSO$_4$; 2 mM CaCl$_2$; pH 7.3) 14 days after transfection using a Nikon Spectral C1si confocal microscope with a NIR Apo ×40 W objective. Z-stacks were made of 30 μm sections using EZ-C1 software (Nikon).

**Immunofluorescence in dissociated neuronal cultures**. Dissociated rat hippocampal neurons (embryonic day 20 (E20)) were sparsely transfected at DIV4–DIV7 with Lipofectamine 2000 (Life Technologies). Briefly, 0.2 μL of Lipofectamine and 0.2 μg of DNA (CRISPR_CaMKIIα and CRISPR_CaMKIIβ) were mixed in 50 μL of Opti-mem (Life Technologies) per well following the manufacturer's guidelines. Neurons were then incubated in Opti-mem transfection medium for 2–3 h. Transfection medium was aspirated, followed by 1× rinse with warm PBS and replacement with fresh warm conditioned medium, then blocked and permeablized in 3% BSA in PBS containing 0.1% Triton-X and stained with primary antibody (2 μg/mL) against CaMKIIα (Enzo Life Sciences) and CaMKIIβ (Abcam) followed by washes in PBSTx and staining with Alexa 488-conjugated secondary (4 μg/mL, Life Technologies A-11034). Covers were mounted in SlowFade Gold (Life Technologies) for imaging. Images were acquired with a ×40 objective and processed and reconstructed using supplied software (NIS-Elements, Nikon). ImageJ was used for post-hoc images editing.

**Immunofluorescence microscopy**. Surface receptors were analyzed using a fluorescence-based antibody binding assay, as previously described[80]. Briefly, hippocampal neurons were transduced with DKO virus at DIV 3 and transfected at DIV 13 with GFP-GluN2B and, where indicated, with CaMKIIα rescue constructs. Transfection was performed using Lipofectamine 2000 (Invitrogen), according to the manufacturer's instructions. At DIV17, surface receptors were labeled with anti-GFP antibody for 15 min at room temperature, fixed with 4% PFA/4% sucrose in PBS and stained with Alexa 488-conjugated secondary antibody (shown in green). After permeabilization with 0.25% Triton X-100 in PBS, the intracellular pool of receptors was labeled with anti-GFP antibody (2 μg/mL, Life Technologies A-11122) followed by Alexa 647-conjugated secondary antibody incubation.

Images were collected using a Zeiss LSM 510 confocal microscope. Serial optical sections of 0.35 μm intervals were used to create maximum projection images. Fluorescence intensity of three independent areas per neuron was analyzed using MetaMorph 6.0 software (Universal Imaging Corp). Intensity is presented as mean ± SEM of the ratio of surface/total intensity. Three to five independent experiments were conducted and significance was analyzed using one-way ANOVA analysis (n = number of cells).

**Statistical analysis**. Data analysis was carried out in Igor Pro (Wavemetrics), Excel (Microsoft), and GraphPad Prism (GraphPad Software). Statistical analyses were performed using the two-tailed Wilcoxon signed-rank test for all experiments using paired whole-cell data, including all synaptic replacement and synaptic overexpression data. In order to compare the different replacements to control and to each mutated form we applied the one-way ANOVA. Normalization was obtained by dividing both control and each condition to the value of the average of the control. Once significance with ANOVA was detected, we applied the Mann

−Whitney U test to calculate the p value. LTP data were gathered from pairs of control and experimental neurons; however, some cells were lost during the experiment. Consequently the resulting datasets are a mix of interleaved and paired data; thus, comparisons were made using the Mann−Whitney test. Summarized data were presented in figures as mean + SEM. In all experiments * is referred to Mann−Whitney p values as follows: ***$p < 0.0001$; **$p < 0.001$; *$p < 0.05$.

**Data availability**. The data that support the findings of this study are available from the corresponding authors upon reasonable request.

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

## Acknowledgements

We thank Dan Qin and Manuel Cerpas for excellent technical assistance and the members of the R.A.N. laboratory for helpful feedback and comments on the manuscript. The authors thank John Lisman and Johannes Hell for constructive discussions. This work was funded by grants from the National Institutes of Mental Health to R.A.N., the NINDS Intramural Research Program to K.W.R. and the Portuguese Foundation for Science and Technology (FCT—Fundacao para a Ciencia e a Tecnologia, Grant SFRH/BI/106010/2015) to M.V. All primary data are archived in the Department of Cellular and Molecular Pharmacology, University of California, San Francisco.

## Author contributions

S.I. performed all electrophysiological, two-photon and spine morphology experiments and all analyses. J.D.-A. performed in utero electroporation surgeries and immuno-fluorescence experiments in Fig. 1. J.I. performed viral injections in adult mice. M.V. and K.W.R. designed and performed the immunofluorescence microscopy experiments. C.S.A. contributed in the design of the gRNAs and performed western blots. K.J.B. contributed in the design of two-photon experiments. V.S.S. contributed in the design and development of adult mice experiments. R.A.N. and S.I. conceived the project and designed the experiments. R.A.N., S.I., C.S.A. and K.J.B. wrote the manuscript. All authors read and edited the paper.

## Additional information

**Competing interests:** The authors declare no competing interests.

