## [Peer Review File · Nature Communications]

Reviewers' comments:

Reviewer #1 (Remarks to the Author):

The manuscript of Incontro et al revisits the long studied role of CaMKII in synaptic plasticity. The authors combined CRISPR/CAS9 knockdown of camkii alpha and beta subunits with rescue transfection of WT and mutated subunits in organotypic slices and slices from animals transfected by in utero electroporation, in order to assess the contribution and mechanism of action of alpha and beta camkii in synaptic transmission and LTP.

The conclusions drawn from this study are largely confirmatory of a wealth of work performed for nearly 30 years., i.e that CaMKII alpha and beta are essential in the induction of LTP, that the interaction of CaMKII with GluN2B is also essential for LTP induction. Some findings, such as the role of CaMKII in basal neurotransmission, had been suggested, but was indeed less recognized. The main novelties in this manuscript are that CaMKII promotes NMDAR-mediated transmission, that CaMKII-GluN2B interaction is also important for the role of CaMKII in supporting basal transmission, and that mainly alpha CaMKII is required for normal synaptic transmission. The results are potentially interesting, although some critical controls, experiments and quantifications of transfection outcome must be made first to validate the conclusions.

Main issues

1- The authors first tried to discriminate the ability of CaMKII alpha vs beta to regulate synaptic transmission. CaMKII is a multimeric enzyme in which both alpha and beta (and possibly other subtypes) co-assemble. Thus, eliminating one subtype with CRISPR/Cas9, such as alpha or beta, should reduce the total number of multimeric CaMKII differently, since the expression levels of each subunit is different. As the authors state in the introduction, aCaMKII has been shown to be more abundant than bCaMKII in the PSD. Consequently, eliminating aCaMKII in neurons is reducing the total amount of CaMKII more dramatically compared to eliminating bCaMKII. Thus, the conclusion from figure 1 that bCaMKII is not required for basal transmission is misleading. It might simply be that reducing total CaMKII by a small proportion (since bCaMKII is not as abundant) has little impact on synaptic transmission. In fact, the double KO results reinforce this point that absolute CaMKII levels may be more relevant here than the specific subtypes in regulating basal synaptic transmission. The partial recovery of synaptic transmission by the bCaMKII rescue (Fig 3) argues that bCaMKII can also contribute to supporting synaptic transmission (as also shown by Okamoto et al, 2015). Knocking down only one subtype and replacing it with the other would be more informative (but see next point).

2- Related to this issue of total CaMKII levels, the authors use various "rescue" combinations by over-expressing back a or bCaMKII or various mutants. However, the authors do not show whether the expression of these constructs lead to comparable and/or different total levels of CaMKII, shedding uncertainties on the interpretations. For instance, the bCaMKII cDNA might not express as well as the aCaMKII; the fluorescence from GFP in the IRES vector cannot accurately report for the abundance of the other gene product (and its stability). A good example of this problem is the comparison between figs 2e and 3j, where knocking out bCaMKII prevents LTP induction (2e), whereas adding back aCaMKII in

the DKO, rescues LTP (3j). If the CRISPR/Cas9+transfection system worked as intended, the two treatments should have yielded the same result on LTP induction. Instead, the results are contradictory, as to the role of bCaMKII, but since we don't know how the total amount of functional CaMKII compares between these two conditions, the results are inconclusive as to the respective contribution of the specific subunits. Also, bCaMKII has several splice variants. Knocking down the gene eliminates all variants, but the cDNA transfection only expresses one subtype. The authors do not address this. In fact, a critical control is missing in the study: CRISPR-Cas9 knockdown of bCaMKII combined with transfection of WT bCaMKII to show complete recovery of LTP, and thus show that this transfected construct is fully functional. Without this control, these data are not interpretable.

3- The authors chose to use a double knock out strategy for the rest of the study while replacing only CaMKII alpha and mutants. Again, differential expression and or stability of different mutants (which have been observed for CaMKII mutants) can confound the interpretations. The authors need to show that each of these mutants are expressing properly to similar levels than endogenous aCaMKII. That said, given the unclear contribution of bCaMKII in synaptic transmission and LTP presented here, the lack of bCaMKII in these experiments may bias the interpretation.

4- The authors used the I205K mutation to show that aCaMKII binding to GluN2B is required for LTP and basal synaptic transmission. However, this mutation in the T-site of the kinase blocks CaMKII interaction with several other binding partners, including itself, microtubules, and presumably many more Ca/CaM-dependent partners (reviewed and analysed in Nguyen et al, Biophysical journal, 2015). Thus, while previous work using GluN2B mutation on the c-tail provided key evidence for a need for this interaction in LTP, the current work is only supportive (consistent with) of this conclusion. To show that this interaction is required in synaptic transmission, the authors would need to knockdown GluN2B and replace it with a mutant impaired in CaMKII binding.

5- The rationale for investigating the structural/catalytic role of aCaMKII in a background without bCaMKII, which was shown to stabilize actin and support spine plasticity via a structural rather than a catalytic role, should be clarified. Do the authors imply that bCaMKII plays no role in sLTP? This imaging protocol would be a good opportunity to repeat bCaMKII knock out and replacement with WT bCaMKII to compare with Okamoto et al (which had different results on the impact on synaptic transmission).

Minor points:

-Clarify time of in utero electroporation. Methods indicate E15.5, but the illustrations on the graph seem to indicate E14

-provide "n" for spine imaging data

-plotting the APV-insensitive run up in all of the LTP graphs is somewhat misleading since it's the same plot in all graphs (not a specific control for each of the concerned experiment).

-Statistics: the authors do not state whether they tested for normality of the distribution

before applying t-tests. Exact p values should be shown for all statistical and non-statistical differences, throughout the manuscript.

-The statement in the abstract: "we conclude that CaMKII fully accounts for NMDAR-dependent LTP", is odd. Other kinases do not play a role? This study did not address this. The statement should be removed (and similar statements in the intro). In the following sentence, the statement "all aspects of CaMKII signaling" is also overstated, since many aspects of CaMKII signaling have not been addressed in this study. These suggestions apply also to the discussion where similar statements are made.

-2nd sentence (intro): I believe the term "isomer" is not appropriate in this sentence. The authors refer to subunits (identical or similar), but not to isomers.

-Consider revising manuscript for English. Examples of desirable revisions: "Ca²⁺ influx...binds..(50)", "autophosphorylates itself (52)", "activity translocates (363)"

Reviewer #2 (Remarks to the Author):

This manuscript reports on a series of well-designed and executed experiments that represent something of a technical tour-de-force. The CRISPR-Cas9 system is used to knockout CaMKIIalpha, CaMKIIbeta or both isoforms in utero, in an in vitro slice culture system and in vivo. As is typical of this group, a series of careful electrophysiological experiments show that this disrupts baseline synaptic transmission mediated by AMPA receptors and NMDA receptors, as well as LTP induction. The authors then test for rescue of these changes by re-expression of a series of mutated CaMKIIalpha proteins, focusing on mutations that have been previously shown to disrupt specific biochemical properties of CaMKII. The impact of many of these mutated proteins has been previously tested in knock-in mice and/or in a variety of in vitro culture systems. In general, the data presented in the manuscript are remarkably "clean", and the authors offer typically robust interpretations that solidify and extend current thinking about the molecular mechanisms underlying LTP. However, there are several issues that require attention:

1. The authors should make more effort to reconcile (within the main body of the text) why they see changes in baseline AMPAR and NMDA transmission that were not reported previously, and why their disruptions of LTP induction data are "cleaner" and more complete than in the prior studies. For example, most prior studies of CaMKIIalpha knockout mice report that NMDAR-dependent LTP is largely, but not completely ablated (as noted in the introduction to this manuscript), in contrast to the complete lack of LTP in the present studies. Such a discussion would be of value in allowing other investigators to understand the (presumably technical?) differences that give rise to the disparity and understand issues related to reproducibility. Similarly, GluN2B knock-in mice that cannot bind CaMKII have only a partial loss of LTP, whereas the I205K mutation completely disrupts LTP. For example, it might be argued that I205K-sensitive interactions with other proteins besides GluN2B (see below) are also important for LTP. Given these prior data, I think the authors

need to soften their interpretations to some extent; while LTP in these preparations is essentially completely ablated, it seems likely that other mechanisms may contribute to LTP in different situations (e.g., ages, brain regions/cell types, stimulation paradigms).

2. The data indicate that the knockdown of CaMKII reduces basal NMDAR transmission, and that CaMKII binding to GluN2B is important for LTP induction. In addition, they report a trend for a reduction in the weighted decay time for the NMDAR currents following knockdown of only CaMKIIa (Supp. Fig. 1c). The authors need to do further studies to more rigorously examine the contribution of GluN2B-NMDARs to basal synaptic transmission with knockdown of both CaMKIIa and CaMKIIb (since this is their baseline for many later studies). In addition, given demonstrated importance of CaMKII binding to GluN2B in LTP induction, additional studies using ifenprodil or related compounds should also be done to examine the contributions of GluN2B-NMDARs, based on NMDAR current amplitudes and decay times. There are reports that CaMKII can directly modulate NMDARs and a previous study indicated that synaptic GluN2B-NMDAR transmission was affected in mice with the CaMKIIa Thr286 to Ala mutation (Gustin et al., 2011).

3. The authors do not comment on the relative expression levels for the various proteins used to rescue the effects of the CRISPR knockout. The interpretations rest to some extent on the assumption that they are all expressed at similar levels. What is the evidence supporting this assumption?

4. The authors need to clean up their verbal descriptions of the effect of the CaMKII mutants that they use. Most critically, they repeatedly refer to the T286A mutant as “kinase dead”, but this is inaccurate – the kinase activity of the T286A mutant is essentially normal in the presence of Ca²⁺/calmodulin. In addition, while the I205K mutant indeed disrupts binding to GluN2B, it also disrupts binding to the CaMKII inhibitor protein (Bayer lab) and to densin (Colbran lab). Indeed, the specificity of its effects on protein-protein interactions are poorly understood. Finally, it is not strictly correct to state for CaMKII that “substrate access to its binding site in the catalytic domain is blocked by the autoinhibitory pseudosubstrate of the protein”.

5. The authors indicate that the only the Students’s t-test is used for comparing differences between data sets. However, many of the studies contain more than 2 groups, making this test inadequate for the task. ANOVA results are needed in several instances.

6. The images of cultured neurons shown in Fig. 1a are too small to be of value.

[Editorial Note: During the first round of review, Reviewer #3 submitted remarks to the editor, which were then relayed to the authors with the reviewer's permission.]

Response to the reviewers' comments

We greatly appreciate the many constructive comments that the reviewers have made. In addressing their concerns we have carried out many additional experiments, some requested and some that we thought of, that we feel greatly improve the manuscript. Below we make a point-by-point response to the issues brought up by the reviewers. All of the changes to the manuscript are highlighted in yellow.

Reviewer #1 (Remarks to the Author):

The manuscript of Incontro et al revisits the long studied role of CaMKII in synaptic plasticity. The authors combined CRISPR/CAS9 knockdown of camkii alpha and beta subunits with rescue transfection of WT and mutated subunits in organotypic slices and slices from animals transfected by in utero electroporation, in order to assess the contribution and mechanism of action of alpha and beta camkii in synaptic transmission and LTP.

The conclusions drawn from this study are largely confirmatory of a wealth of work performed for nearly 30 years., i.e that CaMKII alpha and beta are essential in the induction of LTP, that the interaction of CaMKII with GluN2B is also essential for LTP induction. Some findings, such as the role of CaMKII in basal neurotransmission, had been suggested, but was indeed less recognized. The main novelties in this manuscript are that CaMKII promotes NMDAR-mediated transmission, that CaMKII-GluN2B interaction is also important for the role of CaMKII in supporting basal transmission, and that mainly alpha CaMKII is required for normal synaptic transmission. The results are potentially interesting, although some critical controls, experiments and quantifications of transfection outcome must be made first to validate the conclusions.

We entirely agree with the overall assessment of our study. Indeed, while our study revisits much of the previous literature on CaMKII, our study is, in our opinion, a valuable addition to the field for at least two reasons. First, by using CRISPR to “acutely” delete and replace mutant forms of CaMKII, we provide a much “cleaner” dissection of the role of CaMKII. Second, by examining the many properties of CaMKII with the same manipulations and from the same lab, an overall unified and coherent picture of the central role of CaMKII in excitatory synapse function emerges.

Main issues

1- The authors first tried to discriminate the ability of CaMKII alpha vs beta to regulate synaptic transmission. CaMKII is a multimeric enzyme in which both alpha and beta (and possibly other subtypes) co-assemble. Thus, eliminating one subtype with CRISPR/Cas9, such as alpha or beta, should reduce the total number of multimeric CaMKII differently, since the expression levels of each subunit is different. As the authors state in the introduction, aCaMKII has been shown to be more abundant than bCaMKII in the PSD. Consequently, eliminating aCaMKII in neurons is reducing the total amount of CaMKII more dramatically compared to eliminating bCaMKII. Thus, the conclusion from figure 1 that bCaMKII is not required for basal transmission is misleading. It might simply be that reducing total CaMKII by a small proportion (since bCaMKII is not as abundant) has little impact on synaptic transmission. In fact, the double KO results reinforce this point that absolute CaMKII levels may be more relevant here than the specific subtypes in regulating basal synaptic transmission. The partial recovery of synaptic transmission by the bCaMKII rescue (Fig 3) argues that bCaMKII can also contribute to supporting synaptic transmission (as also shown by Okamoto et al, 2015). Knocking down only one subtype and replacing it with the other would be more informative (but see next point).

Our statement that bCaMKII is not required for basal transmission was not intended to imply anything more than when bCaMKII is deleted basal transmission is unaltered. There are many

reasons for why this might be so. First, bCaMKII is present at low levels, as mentioned by us and the reviewer. Second, bCaMKII maybe intrinsically less effective. This is what we address in later experiments. We conclude in the Discussion that both factors likely contribute to our initial finding. The results from replacement experiments indicate that bCaMKII is not as effective as aCaMKII in maintaining basal transmission. In addition, our new experiment (suggested by the this reviewer) (Fig. S3b) shows that we are unable to rescue the loss of LTP following the deletion of aCaMKII, by expressing bCaMKII (see page 7).

We have reworded and modified our statement in the Discussion concerning the lack of effect of deleting bCaMKII (see page 17).

2- Related to this issue of total CaMKII levels, the authors use various “rescue” combinations by over-expressing back a or bCaMKII or various mutants. However, the authors do not show whether the expression of these constructs lead to comparable and/or different total levels of CaMKII, shedding uncertainties on the interpretations. For instance, the bCaMKII cDNA might not express as well as the aCaMKII; the fluorescence from GFP in the IRES vector cannot accurately report for the abundance of the other gene product (and its stability). A good example of this problem is the comparison between figs 2e and 3j, where knocking out bCaMKII prevents LTP induction (2e), whereas adding back aCaMKII in the DKO, recues LTP (3j). If the CRISPR/Cas9+transfection system worked as intended, the two treatments should have yielded the same result on LTP induction. Instead, the results are contradictory, as to the role of bCaMKII, but since we don’t know how the total amount of functional CaMKII compares between these two conditions, the results are inconclusive as to the respective contribution of the specific subunits. Also, bCaMKII has several splice variants. Knocking down the gene eliminates all variants, but the cDNA transfection only expresses one subtype. The authors do not address this. In fact, a critical control is missing in the study: CRISPR-Cas9 knockdown of bCaMKII combined with transfection of WT bCaMKII to show complete recovery of LTP, and thus show that this transfected construct is fully functional. Without this control, these data are not interpretable.

There are a couple of distinct points contained in this comment. First concerns the relative expression levels in our rescue experiments and whether this could explain some of our findings. The reviewer is correct that we do not have a way to determine the exact levels of the protein we are expressing. This is one of the limitations of using sparse transfections. We would argue that since bCaMKII can fully rescue LTP when bCaMKII is deleted indicates that it is expressed at adequate amounts. Furthermore, we have carried out numerous experiments over the years comparing the effects of expressing numerous proteins before and after an IRES. We invariably find that the effects are dramatically more robust when the expression is before an IRES. Thus, when expressing GFP after the IRES, we feel confident that the protein is being expressed robustly. We do not feel that results between Fig. 2e and Fig. 3j are necessarily contradictory. One explanation for the results in Fig. 2e is that the deletion of bCaMKII reduces the total amount of CaMKII sufficiently to abrogate LTP. Because bCaMKII cannot rescue LTP in the aCaMKII KO (Fig. S3), the effect in Fig. 2e presumably is due to the loss of aCaMKII. (The lack of effect of deleting bCaMKII on basal transmission (Fig. 2d) suggests that it is less sensitive to the deletion of bCaMKII). The full rescue of LTP by aCaMKII indicates that aCaMKII is the key subunit responsible for LTP.

The reviewer raises an interesting point concerning the fact that there are several splice variants of bCaMKII and suggests an experiment to address this issue. We have now carried out this experiment and find that the expression of bCaMKII can fully rescue the loss of LTP when bCaMKII is deleted. We thank the reviewer of this suggestion (see page 7).

3- The authors chose to use a double knock out strategy for the rest of the study while replacing only CaMKII alpha and mutants. Again, differential expression and or stability of different mutants (which have been observed for CaMKII mutants) can confound the interpretations. The authors need to show that each of these mutants are expressing properly to similar levels than endogenous aCaMKII. That said, given the unclear contribution of bCaMKII in synaptic transmission and LTP presented here, the lack of bCaMKII in these experiments may bias the interpretation.

As discussed in response to the reviewer's first point, we cannot precisely determine the expression level of our individual constructs. This is a limitation of our strategy. However, we feel confident that each of the constructs is being adequately expressed because we have the rescue of the NMDAR EPSC as a control, which we now explicitly mentioned (see page 8).

4- The authors used the I205K mutation to show that aCaMKII binding to GluN2B is required for LTP and basal synaptic transmission. However, this mutation in the T-site of the kinase blocks CaMKII interaction with several other binding partners, including itself, microtubules, and presumably many more Ca/CaM-dependent partners (reviewed and analysed in Nguyen et al, Biophysical journal, 2015). Thus, while previous work using GluN2B mutation on the c-tail provided key evidence for a need for this interaction in LTP, the current work is only supportive (consistent with) of this conclusion. To show that this interaction is required in synaptic transmission, the authors would need to knockdown GluN2B and replace it with a mutant impaired in CaMKII binding.

We agree with the concerns raised by the reviewer and have spent most of our time addressing this point (see pages 12-13). First, we have now carried out experiments in which we disrupt binding by making a mutation at the S-site. This mutation is as effective as the T-site mutation in disrupting the synaptic effects of CaMKII. The reviewer is correct that these mutations would also affect the binding of CaMKII to other proteins such as densin. We would conclude from our results with the CaMKII mutants that CaMKII must bind to synaptic proteins in order to have any effect (except the rescue of NMDAR currents) on synaptic function. As the reviewer suggests the reciprocal mutation in the GluN2B C-tail is required to determine the relative importance of this interaction in the synaptic actions of CaMKII. At his/her suggestion we have now carried out these experiments, which have been most informative (see new Figure 8). First, KO of GluN2B causes a dramatic reduction in the NMDAR EPSCs and an enhancement of AMPAR EPSCs, as expected from previous results (Gray et al., Neuron, 2011). We then carried out a rescue by expressing a mutation in the GluN2B C-tail that abolishes the binding of CaMKII. This construct fully rescued the NMDAR EPSC, demonstrating that this mutation had no effect on the function of the NMDAR. Importantly, the AMPAR EPSCs are reduced to an extent similar to that seen with a kinase inactive mutant of CaMKII. This finding indicates that the binding of CaMKII to the NMDAR is necessary for its role in maintaining basal synaptic transmission. Finally, and most importantly, the dramatic enhancement of synaptic transmission observed with the kinase active T305A/T306A mutations is absent. Together these findings establish that for CaMKII to have any effect on synaptic function, except for its effects on NMDARs, it must be docked at the PSD by its binding to the NMDAR. In addition to the additions to the results section, we also include these findings in the Discussion (see page 21).

5- The rationale for investigating the structural/catalytic role of aCaMKII in a background without bCaMKII, which was shown to stabilize actin and support spine plasticity via a structural rather than a catalytic role, should be clarified. Do the authors imply that bCaMKII plays no role in sLTP? This imaging protocol would be a good opportunity to repeat bCaMKII knock out and

replacement with WT bCaMKII to compare with Okamoto et al (which had different results on the impact on synaptic transmission).

This is another good point raised by the reviewer. We have now examined the effect of deleting bCaMKII on sLTP experiments. As illustrated in Fig. S6g and j, deleting bCaMKII blocks sLTP (see pages 9-10).

Minor points:

-Clarify time of in utero electroporation. Methods indicate E15.5, but the illustrations on the graph seem to indicate E14

This has been remedied.

-provide “n” for spine imaging data

Done.

-plotting the APV-insensitive run up in all of the LTP graphs is somewhat misleading since it's the same plot in all graphs (not a specific control for each of the concerned experiment).

Throughout the LTP experiments we were concerned about the stability of the baseline. Because of the “washout” of LTP with whole cell recording, we were unable to obtain a sufficiently long baseline to ensure its stability throughout the experiment. Thus during the course of these experiments we interleaved experiments in which we applied APV. Thus we feel confident that the results with APV accurately reflect, on average, the baseline for the LTP experiments. We now expand on this (see page 6).

-Statistics: the authors do not state whether they tested for normality of the distribution before applying t-tests. Exact p values should be shown for all statistical and non-statistical differences, throughout the manuscript.

We have provided the requested information.

-The statement in the abstract: “we conclude that CaMKII fully accounts for NMDAR-dependent LTP”, is odd. Other kinases do not play a role? This study did not address this. The statement should be removed (and similar statements in the intro). In the following sentence, the statement “all aspects of CaMKII signaling” is also overstated, since many aspects of CaMKII signaling have not been addressed in this study. These suggestions apply also to the discussion where similar statements are made.

The reviewer is correct. This paper provides no information on the sufficiency of CaMKII for LTP. However, previous results with constitutively active CaMKII (Lledo et al., PNAS, 1995; Petit et al., Science, 1994) have shown that it faithfully mimics and occludes LTP. Thus we see no reason to implicate other kinases in LTP. We have modified the abstract and introduction accordingly. We do elaborate in the Discussion on the necessity and sufficiency of CaMKII in LTP. In addition we have changed “all aspects of CaMKII signaling” to “all aspect of CaMKII signaling examined”.

-2nd sentence (intro): I believe the term “isomer” is not appropriate in this sentence. The authors refer to subunits (identical or similar), but not to isomers.

Point well taken. We have changed “isomer” to “subunit” throughout the text.

-Consider revising manuscript for English. Examples of desirable revisions: “Ca²⁺ influx...binds..(50)”, “autophosphorylates itself (52)”, “activity translocates (363)”

We have made the requested changes.

Reviewer #2 (Remarks to the Author):

This manuscript reports on a series of well-designed and executed experiments that represent something of a technical tour-de-force. The CRISPR-Cas9 system is used to knockout CaMKIIalpha, CaMKIIbeta or both isoforms in utero, in an in vitro slice culture system and in vivo. As is typical of this group, a series of careful electrophysiological experiments show that this disrupts baseline synaptic transmission mediated by AMPA receptors and NMDA receptors, as well as LTP induction. The authors then test for rescue of these changes by re-expression of a series of mutated CaMKIIalpha proteins, focusing on mutations that have been previously shown to disrupt specific biochemical properties of CaMKII. The impact of many of these mutated proteins has been previously tested in knock-in mice and/or in a variety of in vitro culture systems. In general, the data presented in the manuscript are remarkably “clean”, and the authors offer typically robust interpretations that solidify and extend current thinking about the molecular mechanisms underlying LTP. However, there are several issues that require attention:

We thank the reviewer for his positive assessment of the contributions of our manuscript.

1. The authors should make more effort to reconcile (within the main body of the text) why they see changes in baseline AMPAR and NMDA transmission that were not reported previously, and why their disruptions of LTP induction data are “cleaner” and more complete than in the prior studies. For example, most prior studies of CaMKIIalpha knockout mice report that NMDAR-dependent LTP is largely, but not completely ablated (as noted in the introduction to this manuscript), in contrast to the complete lack of LTP in the present studies. Such a discussion would be of value in allowing other investigators to understand the (presumably technical?) differences that give rise to the disparity and understand issues related to reproducibility. Similarly, GluN2B knock-in mice that cannot bind CaMKII have only a partial loss of LTP, whereas the I205K mutation completely disrupts LTP. For example, it might be argued that I205K-sensitive interactions with other proteins besides GluN2B (see below) are also important for LTP. Given these prior data, I think the authors need to soften their interpretations to some extent; while LTP in these preparations is essentially completely ablated, it seems likely that other mechanisms may contribute to LTP in different situations (e.g., ages, brain regions/cell types, stimulation paradigms).

The reviewer brings up an issue that we have wrestled with throughout this study. There are many findings in our study that are either quantitatively or even qualitatively different from previous findings. An obvious and well-used explanation for germline KOs is compensation. To be honest, in terms of CaMKII we have no idea what this would mean. It is most unlikely that CaMKII beta could compensate, because it is unable to support LTP. The result that most directly conflicts with our results is the conditional KO of CaMKIIalpha in the adult hippocampus (Achterberg et al., J. Neurosci.2014) where they find no effect on baseline transmission. We, on the other hand, find a dramatic reduction in baseline transmission following CRISPR KO of CaMKIIalpha and beta in the adult (Fig. S4). We would argue that our paired recording approach is more sensitive than the use of field potentials recorded across slices, as used in the Achterberg paper. We now devote a separate paragraph near the beginning of the Discussion (second paragraph) explicitly

addressing these issues (see page 16).

2. The data indicate that the knockdown of CaMKII reduces basal NMDAR transmission, and that CaMKII binding to GluN2B is important for LTP induction. In addition, they report a trend for a reduction in the weighted decay time for the NMDAR currents following knockdown of only CaMKIIa (Supp. Fig. 1c). The authors need to do further studies to more rigorously examine the contribution of GluN2B-NMDARs to basal synaptic transmission with knockdown of both CaMKIIa and CaMKIIb (since this is their baseline for many later studies). In addition, given demonstrated importance of CaMKII binding to GluN2B in LTP induction, additional studies using ifenprodil or related compounds should also be done to examine the contributions of GluN2B-NMDARs, based on NMDAR current amplitudes and decay times. There are reports that CaMKII can directly modulate NMDARs and a previous study indicated that synaptic GluN2B-NMDAR transmission was affected in mice with the CaMKIIa Thr286 to Ala mutation (Gustin et al., 2011).

We thank the reviewer for bringing to our attention the Gustin et al. paper. This is an important issue and we have gone back and examined in more detail the kinetics of the NMDAR EPSC, both in the CaMKIIa KO and in the DKO. We could find no change in kinetics following these deletions and we have added this new data to Fig. S1c. Thus we do not feel that there is a change in the subunit composition of the NMDA receptor following the deletion of CaMKII. We now compare our findings to those of Gustin et al. in the Discussion (see page 17, lines 5-7 from the top).

3. The authors do not comment on the relative expression levels for the various proteins used to rescue the effects of the CRISPR knockout. The interpretations rest to some extent on the assumption that they are all expressed at similar levels. What is the evidence supporting this assumption?

This issue was also raised by reviewer #1. As discussed earlier, we are unable to accurately determine the level of expression of the proteins. This is a limitation of the sparse transfection system that we use. However, for virtually all of the experiments in this study we monitor the NMDAR EPSC and show that it is fully rescued in all of our experiments, verifying that the protein is being expressed. We now address this issue in the text.

4. The authors need to clean up their verbal descriptions of the effect of the CaMKII mutants that they use. Most critically, they repeatedly refer to the T286A mutant as “kinase dead”, but this is inaccurate – the kinase activity of the T286A mutant is essentially normal in the presence of Ca²⁺/calmodulin. In addition, while the I205K mutant indeed disrupts binding to GluN2B, it also disrupts binding to the CaMKII inhibitor protein (Bayer lab) and to densin (Colbran lab). Indeed, the specificity of its effects on protein-protein interactions are poorly understood. Finally, it is not strictly correct to state for CaMKII that “substrate access to its binding site in the catalytic domain is blocked by the autoinhibitory pseudosubstrate of the protein”.

We thank the reviewer for correcting us on the nomenclature. We now refer to the T286A mutation as “inactive”, throughout the manuscript. The issues with the I205K mutation was also mentioned by Reviewer #1 and we have modified the description of this mutant and the effects that we observe. Most importantly we have carried out the reciprocal mutations in the GluN2B C-tail to determine how important the disruption of NMDAR binding is to the overall effects of the I205K results. Remarkably the deficits in synaptic transmission observed with the mutant GluN2B are identical to that seen with the kinase dead CaMKII mutants. Finally, the pronounced enhancement in transmission seen with the T305A/T306A mutant is absent when CaMKII cannot

bind to the modified GluN2B mutant.

5. The authors indicate that the only the Students's t-test is used for comparing differences between data sets. However, many of the studies contain more than 2 groups, making this test inadequate for the task. ANOVA results are needed in several instances.

We have now applied ANOVA to all the experimental groups. We also made a post-hoc analysis using Mann-Whitney test to calculate p-values and indicated them in the figures legends. As the reviewer correctly requests we have now indicated the significance in between the experimental conditions, not only in respect to the control.

6. The images of cultured neurons shown in Fig. 1a are too small to be of value.

We have replaced these images with the immunofluorescence images of CaMKII.

Reviewer #3, as relayed by the editor. Reviewer#3 requests imaging of CaMKII to verify the CRISPR KO. We have now carried out the experiments in dissociated neuronal cultures and show that CaMKII α and CaMKII β are absent in cells expressing Cas9 and the guide RNAs. This is now Fig. 1c and d.

We thank the reviewers for their many detailed and most constructive suggestions. We have found them enormously helpful and have incorporated virtually all of their suggestions into the modified manuscript.

Reviewers' comments:

Reviewer #1 (Remarks to the Author):

The authors have made extensive revisions to their manuscripts which improve the study significantly. I have no major concern.

Only two minors points:

Throughout the paper, the authors now refer to T286A mutant as a "kinase inactive mutant" (as opposed to "dead"). This is still not correct. This mutant is active upon Ca/CaM activation (but cannot autophosphorylate at T286 obviously nor become "autonomous" once Ca drops back down). It differs from K42R which is truly inactive, even upon Ca/CaM activation.

The title of suppl fig 8 should be reworded ("Neurons expressing CaMKIIa K42R shows residual catalytic activity"). Other than a grammatical error, the title is not actually backed up with a measurement of "catalytic activity". The title should describe more directly the observed results (not the interpretation of residual catalytic activity, which is speculative here).

Reviewer #2 (Remarks to the Author):

The authors have added new data and made several changes to the manuscript. However, I am not convinced that these changes have substantially improved the manuscript. I have addressed the authors' responses to the prior concerns in turn, and then mentioned some additional relatively minor issues.

Prior Point 1. Contrast with prior studies. I do not find the new 4 line paragraph added to Page 16 to be very helpful or satisfying. Mainly, I disagree that the CRISPR approach used in most of these studies is different and superior because it is an "acute" deletion/rescue relative to the various transgenic lines studied previously. This is especially the case for the in utero and in vitro studies. I do not understand how in utero CRISPR deletion at E15, well before CaMKIIalpha is normally expressed at around post-natal day 5, can be any different from a germline null mouse when analyses are conducted at P20-28. Similarly, the effects of in vitro CRISPR deletion at 2DIV are analyzed at DIV12-14. The studies in adult mice shown in Fig. S4 are better in this regard, but they are not shown in the main manuscript. Most certainly none of these manipulations can be considered "acute" deletions. In fact, if I understand correctly, the expression of rescue proteins is being driven by non-native promoters, potentially introducing artefactual effects to the in utero and in vitro studies due to premature protein expression. Thus, differences between these findings and the prior studies remain perplexing to me, despite the revisions and the authors rebuttal.

As I reflected on these issues, I considered that the pairing LTP protocol (0 mV, 2 Hz/90 s)

used here is somewhat different from the perhaps more widely used theta burst or 100 Hz protocols in prior slice studies. Could this explain the apparent contributions of CaMKII-independent mechanisms in prior studies? Is the global block of LTP after the CRISPR KO of alpha (or the DKO) also observed using theta burst or 100 Hz protocols? One might posit the reduction of AMPA EPSC amplitudes in the KO cells means that this less intensive stimulation used in the current studies is insufficient to provide sufficient depolarization to release the Mg²⁺ block of the NMDA receptors, resulting in a lack of LTP, even when the NMDA EPSC is rescued (as in Fig. 5). A stronger stimulation might overcome such a deficit.

Prior Point 2. NMDAR current analyses. Thanks for addressing this issue. However, the exemplar traces shown in these new data indicate that the NMDAR EPSCs in CRISPR_GluN2B cells (Fig. 8g) have much slower kinetics than control, and that this effect is rescued by re-expressing the mutated GluN2B, as was the amplitude (Figs. 8 h/i). However, this is opposite to what one would expect if GluN2B were deleted. Why? Please report the average NMDAR EPSC kinetics for this figure, as in supplementary figure 1c.

Prior Point 3. Expression levels. This becomes more complex when comparing the effects of expressing mutated forms of the same protein when the mutation may have an impact on the stability of the protein resulting in changes in steady state proteins levels (i.e., CaMKIIalpha mutants in most cases here). Thus, the confounding effects of variations in protein expression levels are not really addressed.

Prior Point 4. Kinase mutant nomenclature. The edits to the text are inadequate. Referring to the T286A mutant as inactive is not accurate. This mutant has normal kinase activity in the presence of calcium/calmodulin. The authors should be commended by addressing specificity issues with the I205K mutant by adding new data in Figure 8 studying a GluN2B mutant previously shown to be deficient in CaMKII binding. However, these findings are hard to interpret because there is no control rescue with WT GluN2B. Does a WT rescue restore AMPAR EPSCs to normal, or to the reduced levels seen with the mutant? Is there any difference in the restoration of NMDAR EPSCs by WT or mutant GluN2B? The statement (p13, lines 4-6) that "these findings establish that for CaMKII to have any effect on synaptic function, except for its effects on NMDARs, it must be docked at the PSD by its binding to the NMDAR" is simply not justified without this control, and especially when the only synaptic functions measured are AMPA and NMDA EPSCs, with no measures of synaptic plasticity, morphology, or other functions.

Also, related to this, I suggest deleting the data with the triple K42R_T305A/T306A mutation. The interpretation that the K42R mutation has a low residual kinase activity that is enhanced by the T305A/T306A mutations is very weak given the extensive prior data showing K42R is a kinase-dead mutation. Biochemical analyses to validate this statement would be required. Rather, the effect of the T305A/T306A mutations may be mediated (in part) by protein interactions affecting subcellular targeting (see Elgersma et al., Neuron 2002) in the context of K42R mutant.

Prior Point 5. Statistical reporting. Thanks for adding more explicit statements to the figure legends about the tests that were used. However, in several instances these seem like

blanket statements that were inserted in the legend without regard to the tests used in the actual figure in question. For example, for figure 2, where was the indicated ANOVA used? The legends for Figs. 2 and 3 indicate that p values from a Mann Whitney test are shown for the LTP studies, but they seem to be missing. The legends of the supplementary figures also need to be updated as in the main manuscript to more completely describe the statistical tests used.

Prior Point 6. Images of neurons. Thanks for including enlarged images supporting the CRISPR efficacy. However, for Figs. 1c/d, the authors should specify when the neurons were fixed and stained relative to the time of lentiviral infection (unless I missed it). Also, does every neuron in these cultures express CaMKII alpha and CaMKIIbeta? If not, then one might argue that the transfected neurons lacking CaMKII expression in panels c and d are merely non-expressing neurons, not neurons from which CRISPR has effectively deleted the CaMKII isoforms.

Additional minor points:

7. Fig. 2c/e/g. A new statement in the text indicated that the APV control data sets in each panel were interleaved throughout each study in these panels. However, the plots appear very similar when overlaid. Please explicitly clarify whether these are independent data sets in each panel, collected at the same time of the other data, or whether the data plotted in each panel are from a single group of the interleaved control experiments and simply re-drawn for comparison.

8. Consider moving Supplementary Fig. 4 to the main text.

9. Fig. 4: Please clarify whether the DKO data shown in the bar graphs (e/h) are independent data sets, or reproductions of the same data shown in Figs. 1 or 3.

10. P14, line 13-14. "...enhancement caused by the constitutively active T286D-T305A/T306A is also completely prevented by disrupting the binding of CaMKII to NMDARs..." Sentence should be modified to indicate that binding to other proteins is also disrupted by I205K mutation, as in other sections.

11. The title or abstract should indicate that all this work is done in the hippocampus.

Reviewer #1 (Remarks to the Author):

The authors have made extensive revisions to their manuscripts which improve the study significantly. I have no major concern.

We appreciate that this reviewers recognizes the extensive additional work we have done to this study.

Only two minors points:

Throughout the paper, the authors now refer to T286A mutant as a “kinase inactive mutant” (as opposed to “dead”). This is still not correct. This mutant is active upon Ca/CaM activation (but cannot autophosphorylate at T286 obviously nor become “autonomous” once Ca drops back down). It differs from K42R which is truly inactive, even upon Ca/CaM activation.

We thank the reviewer for pointing this out. We now refer to T286A as “autophosphorylation dead”.

The title of suppl fig 8 should be reworded (“Neurons expressing CaMIIa K42R shows residual catalytic activity”). Other than a grammatical error, the title is not actually backed up with a measurement of “catalytic activity”. The title should describe more directly the observed results (not the interpretation of residual catalytic activity, which is speculative here).

The reviewer has a good point. We have change the title.

Reviewer #2 (Remarks to the Author):

The authors have added new data and made several changes to the manuscript. However, I am not convinced that these changes have substantially improved the manuscript. I have addressed the authors' responses to the prior concerns in turn, and then mentioned some additional relatively minor issues.

We were surprised and puzzled by the seeming change in the reviewer's enthusiasm for our study. In the initial review he/she stated: “This manuscript reports on a series of well-designed and executed experiments that represent something of a technical tour-de-force”, “the data presented in the manuscript are remarkably “clean”, and the authors offer typically robust interpretations that solidify and extend current thinking about the molecular mechanisms underlying LTP”. He/she then provided a series of highly constructive and detail suggestions, which we took months to address. Thus we were taken aback by his/her change of tone.

Prior Point 1. Contrast with prior studies. I do not find the new 4 line paragraph

added to Page 16 to be very helpful or satisfying. Mainly, I disagree that the CRISPR approach used in most of these studies is different and superior because it is an "acute" deletion/rescue relative to the various transgenic lines studied previously. This is especially the case for the in utero and in vitro studies. I do not understand how in utero CRISPR deletion at E15, well before CaMKIIalpha is normally expressed at around post-natal day 5, can be any different from a germline null mouse when analyses are conducted at P20-28. Similarly, the effects of in vitro CRISPR deletion at 2DIV are analyzed at DIV12-14. The studies in adult mice shown in Fig. S4 are better in this regard, but they are not shown in the main manuscript. Most certainly none of these manipulations can be considered "acute" deletions. In fact, if I understand correctly, the expression of rescue proteins is being driven by non-native promoters, potentially introducing artefactual effects to the in utero and in vitro studies due to premature protein expression. Thus, differences between these findings and the prior studies remain perplexing to me, despite the revisions and the authors rebuttal.

We agree entirely with the reviewer's comments and we just quite simply do not have a sensible explanation for the different findings. The only obvious difference between our study and most previous studies is that we used single cell manipulations, whereas most previous studies used global manipulations. There are a couple of studies, one on neuroligin (Kwon et al., Nat. Neurosci. 2012) and the other on ephrin-B3 (McClelland et al., PNAS, 2010) that directly compared knock down of neuroligin 1 using either sparse or global strategies. They found that in the global manipulation these synaptic adhesion molecules had no effect on synapse number, whereas single cell manipulations did. They propose that there may be an intercellular competition for binding partners. However, it is not obvious as to how this model might explain our results. We now discuss these possibilities, but would welcome any further input from the reviewer.

As I reflected on these issues, I considered that the pairing LTP protocol (0 mV, 2 Hz/90 s) used here is somewhat different from the perhaps more widely used theta burst or 100 Hz protocols in prior slice studies. Could this explain the apparent contributions of CaMKII-independent mechanisms in prior studies? Is the global block of LTP after the CRISPR KO of alpha (or the DKO) also observed using theta burst or 100 Hz protocols? One might posit the reduction of AMPA EPSC amplitudes in the KO cells means that this less intensive stimulation used in the current studies is insufficient to provide sufficient depolarization to release the Mg²⁺ block of the NMDA receptors, resulting in a lack of LTP, even when the NMDA EPSC is rescued (as in Fig. 5). A stronger stimulation might overcome such a deficit.

There is a long history on various approaches to studying LTP. There are two separate issues concerning LTP. The first involves processes that control whether LTP is induced, i.e., what controls the activation of the NMDAR. This involves a myriad of factors, e.g., the level of inhibition, the level of AMPAR

transmission, neuronal excitability, transmitter release, etc. For instance, if we reduced the AMPAR currents by 70% with NBQX and then induce LTP with a tetanus, LTP will be reduced under this condition, because the cell will not depolarize adequately during the tetanus. However, it would be incorrect to say that AMPARs play a direct role in the mechanism underlying LTP. Indeed, when pairing is used, LTP is entirely normal in this condition.

The second issue involves understanding the actual mechanism of LTP, i.e., what occurs downstream of NMDAR activation. To bypass all of the factors that control the activation of the NMDAR, we use a “pairing protocol”. If the manipulation does not affect the NMDAR current, but does affect LTP, we would conclude that it is disrupting the mechanism underlying LTP. Also because we provide the identical depolarization in all of our experiments, it greatly reduces the variability across experiments.

Since the LTP generated by the tetanus protocols referred to by reviewer are generally reported to be entirely blocked by APV, we would assume that we are looking at the same phenomenon.

Prior Point 2. NMDAR current analyses. Thanks for addressing this issue. However, the exemplar traces shown in these new data indicate that the NMDAR EPSCs in CRISPR_GluN2B cells (Fig. 8g) have much slower kinetics than control, and that this effect is rescued by re-expressing the mutated GluN2B, as was the amplitude (Figs. 8 h/i). However, this is opposite to what one would expect if GluN2B were deleted. Why? Please report the average NMDAR EPSC kinetics for this figure, as in supplementary figure 1c.

We believe that the reviewer may not realize that these experiments were carried out with the AMPARs intact. The experimental trace (green) actually decays considerably faster than the control (black) trace in Fig. 9g. This can be explained by both the decrease in the NMDAR current (measured at 100 ms) and the early enhancement in the response due to the AMPAR enhancement. Because of the changes in the AMPAR EPSC, it is not possible to evaluate the decay kinetics of the pure NMDAR current. We do not feel that repeating all of the experiments in Fig. 9 in the presence of NBQX, would contribute appreciably to the conclusions reached from the data in Fig. 9. As discussed in our response to point #4, we have now added the experiment with the wt rescue. As the reviewer pointed out this is the more appropriate control.

Prior Point 3. Expression levels. This becomes more complex when comparing the effects of expressing mutated forms of the same protein when the mutation may have an impact on the stability of the protein resulting in changes in steady state proteins levels (i.e., CaMKIIalpha mutants in most cases here). Thus, the confounding effects of variations in protein expression levels are not really addressed.

Our original response to point #3 was that with all of our rescue experiments, the

mutated constructs fully rescued the NMDAR. This indicates to us that we have adequately expressed the construct. We feel that this is a strong internal control for our experiments. We are happy to acknowledge that we don't not know the exact proteins levels of the expressed construct.

Prior Point 4. Kinase mutant nomenclature. The edits to the text are inadequate. Referring to the T286A mutant as inactive is not accurate. This mutant has normal kinase activity in the presence of calcium/calmodulin. The authors should be commended by addressing specificity issues with the I205K mutant by adding new data in Figure 8 studying a GluN2B mutant previously shown to be deficient in CaMKII binding. However, these findings are hard to interpret because there is no control rescue with WT GluN2B. Does a WT rescue restore AMPAR EPSCs to normal, or to the reduced levels seen with the mutant? Is there any difference in the restoration of NMDAR EPSCs by WT or mutant GluN2B? The statement (p13, lines 4-6) that "these findings establish that for CaMKII to have any effect on synaptic function, except for its effects on NMDARs, it must be docked at the PSD by its binding to the NMDAR" is simply not justified without this control, and especially when the only synaptic function measured are AMPA and NMDA EPSCs, with no measures of synaptic plasticity, morphology, or other functions.

We appreciate his/her corrections concerning the kinase mutant nomenclature. We now refer to T286A as an autophosphorylation dead mutant. The reviewer brings up a good point about rescuing with wt GluN2B. Although we have previously carried out this experiment in a different context (Sanz-Clemente et al., Cell Reports, 2013) and found that the wt fully rescues NMDAR currents, we did not examined the AMPAR currents. As requested by the reviewer we have carried out the wt rescue experiment. As predicted there is a complete rescue of the NMDAR and no change in the AMPAR responses. We thank the reviewer for this suggestion and have added this data to the figure and text. More specifically, added the experiment as panel d and h of Fig. 9 with the rescue data with wt receptor subunit.

Also, related to this, I suggest deleting the data with the triple K42R_T305A/T306A mutation. The interpretation that the K42R mutation has a low residual kinase activity that is enhanced by the T305A/T306A mutations is very weak given the extensive prior data showing K42R is a kinase-dead mutation. Biochemical analyses to validate this statement would be required. Rather, the effect of the T305A/T306A mutations may be mediated (in part) by protein interactions affecting subcellular targeting (see Elgersma et al., Neuron 2002) in the context of K42R mutant.

The reviewer raises an important point. We agree that the data with the K42R_T305A/T306A is not as definitive as one would like. We have therefore removed this data as requested by the reviewer. We retain the results on basal synaptic transmission and LTP.

Prior Point 5. Statistical reporting. Thanks for adding more explicit statements to the figure legends about the tests that were used. However, in several instances these seem like blanket statements that were inserted in the legend without regard to the tests used in the actual figure in question. For example, for figure 2, where was the indicated ANOVA used? The legends for Figs. 2 and 3 indicate that p values from a Mann Whitney test are shown for the LTP studies, but they seem to be missing. The legends of the supplementary figures also need to be updated as in the main manuscript to more completely describe the statistical tests used.

We thank the reviewer for the constructive suggestions on statistics. We have now added all p values to normalized data and specified which test was used for the respective experiments. We have also added specific information on statistics in the supplementary section.

Prior Point 6. Images of neurons. Thanks for including enlarged images supporting the CRISPR efficacy. However, for Figs. 1c/d, the authors should specify when the neurons were fixed and stained relative to the time of lentiviral infection (unless I missed it). Also, does every neuron in these cultures express CaMKII alpha and CaMKIIbeta? If not, then one might argue that the transfected neurons lacking CaMKII expression in panels c and d are merely non-expressing neurons, not neurons from which CRISPR has effectively deleted the CaMKII isoforms.

We thank the reviewer for raising this possibility. The vast majority of neurons in our culture conditions express CaMKIIalpha and beta. Furthermore the immunofluorescence experiments were done using a lipofectamine protocol not with lentiviral infections. Lentiviral infection was used only for Western blot as indicated in the methods.

Additional minor points:

7. Fig. 2c/e/g. A new statement in the text indicated that the APV control data sets in each panel were interleaved throughout each study in these panels. However, the plots appear very similar when overlaid. Please explicitly clarify whether these are independent data sets in each panel, collected at the same time of the other data, or whether the data plotted in each panel are from a single group of the interleaved control experiments and simply re-drawn for comparison.

Over the period of this study we carried out “control” experiments in the presence of APV. We concluded that on average we had a run up of about 50%. We therefore averaged this data set together and reproduce it in each of the experiments. Short of repeating this APV “controls” for every experiment, which would have double our efforts, we feel that this is best way to present the data. We have been more explicit in the text.

8. Consider moving Supplementary Fig. 4 to the main text.

We would be happy to move this figure to the main text. In the new version of the revised manuscript, we moved it as figure 4.

9. Fig. 4: Please clarify whether the DKO data shown in the bar graphs (e/h) are independent data sets, or reproductions of the same data shown in Figs. 1 or 3.

This is a good point. Indeed, this is the same data as presented earlier, which we include in the graph to make it easy to compare the different results. We now explicitly state this in the figure legend.

10. P14, line 13-14. "...enhancement caused by the constitutively active T286D-T305A/T306A is also completely prevented by disrupting the binding of CaMKII to NMDARs..." Sentence should be modified to indicate that binding to other proteins is also disrupted by I205K mutation, as in other sections.

We have modified the sentence as follows: "enhancement caused by constitutively active CaMKII α T286D-T305A/T306A is also completely prevented by the I205K mutation suggesting that the binding of CaMKII to NMDARs is essential for the actions of CaMKII"

11. The title or abstract should indicate that all this work is done in the hippocampus.

This is a good point.

REVIEWERS' COMMENTS:

Reviewer #1 (Remarks to the Author):

I have not further comments

Reviewer #2 (Remarks to the Author):

I apologize if my second series of comments did not fully convey my continued general enthusiasm for the work (as expressed in the original critique). Thanks for addressing the remaining concerns. I am satisfied with all the responses, with the exception of the description of the T286A CaMKII mutant as "autophosphorylation dead": this is misleading as this mutant can be autophosphorylated at sites other than Thr286. Perhaps "Thr286-autophosphorylation null"?